# Silencing Nfix rescues muscular dystrophy by delaying muscle regeneration

Giuliana Rossi [1], Chiara Bonfanti[1], Stefania Antonini[1], Mattia Bastoni[1], Stefania Monteverde[1], Anna Innocenzi[2], Marielle Saclier[1], Valentina Taglietti[1] & Graziella Messina [1]

Muscular dystrophies are severe disorders due to mutations in structural genes, and are characterized by skeletal muscle wasting, compromised patient mobility, and respiratory functions. Although previous works suggested enhancing regeneration and muscle mass as therapeutic strategies, these led to no long-term benefits in humans. Mice lacking the transcription factor *Nfix* have delayed regeneration and a shift toward an oxidative fiber type. Here, we show that ablating or silencing the transcription factor *Nfix* ameliorates pathology in several forms of muscular dystrophy. Silencing *Nfix* in postnatal dystrophic mice, when the first signs of the disease already occurred, rescues the pathology and, conversely, *Nfix* overexpression in dystrophic muscles increases regeneration and markedly exacerbates the pathology. We therefore offer a proof of principle for a novel therapeutic approach for muscular dystrophies based on delaying muscle regeneration.

[1] Department of Biosciences, University of Milan, via Celoria 26, 20133 Milan, Italy. [2] Division of Regenerative Medicine, Stem Cells and Gene Therapy, San Raffaele Scientific Institute, via Olgettina 60, 20132 Milan, Italy. Correspondence and requests for materials should be addressed to G.M. (email: graziella.messina@unimi.it)

Muscular dystrophies (MDs) are inherited skeletal muscle disorders characterized by progressive muscle damage and weakness of variable distribution and severity, leading to wheelchair dependency and, in the most severe cases, to patient's death[1, 2]. MDs are due to mutations in genes encoding for proteins of the structural dystrophin-glycoprotein complex, which induce sarcolemmal instability and muscle necrosis. The most common form is Duchenne muscular dystrophy (DMD), an X-linked autosomal recessive disorder due to mutations in the dystrophin gene, which encodes a protein anchoring the sarcolemmal membrane to the cytoskeleton, thus protecting the fibers from contraction-induced damage[3]. Another form is limb girdle muscular dystrophy 2D, an autosomal recessive disorder caused by mutations in the α-sarcoglycan gene[4] and part of a group of MDs, with a prevalence ranging from 1 in 14,500 to 1 in 123,000[5, 6].

Unfortunately, there is no effective therapy, and corticosteroids represent the most widely used treatment to counteract chronic inflammation[7]. Although many attempts have been done to design cell and gene therapies, these approaches are limited by technical issues, related to difficulties in finding the appropriate cell type or vector, and are therefore still far to be curative[8, 9]. A shared knowledge in the field is that to be really successful, any therapeutic approach has to rely on good muscle quality, therefore restricting the number of patients eligible for clinical trials. In fact, there is no available approach able to rescue muscle damage when the muscle tissue has been completely lost and substituted by fibrotic deposits, thus restricting the cohort of patients eligible for clinical trials to the youngest and less compromised individuals. Therefore, many groups proposed drugs and genetic constructs to counteract skeletal muscle degeneration by promoting regeneration by endogenous satellite cells[10, 11]. Nevertheless, none of these strategies demonstrated to be efficacious when translated to humans[12], and there is still the need for alternative approaches and new targets to be identified. MDs are indeed characterized by continuous cycles of degeneration and regeneration, as a consequence of the attempt to repair damage by satellite stem cells that unfortunately, sharing the same mutation of the myofibers, are not able to successfully repair damage, leading to the loss of muscle tissue and establishment of fibrosis. Interestingly, it has been shown that slow-twitch, oxidative fibers are more protected from damage-induced oxidative stress and degeneration[13–16]. In light of our recent observation that mice lacking Nfix are characterized by a delayed regeneration after injury and a switch toward a slow-twitch phenotype[17, 18], we hypothesized that targeting Nfix in a dystrophic context may exert a protective effect.

Nfix is part of a family of four closely related transcription factors (Nfia, b, c, and x) with a role in activating/repressing transcription of genes expressed in various organs[19–25]. We previously demonstrated that Nfix is responsible for the transcriptional switch from embryonic to fetal myogenesis, a crucial checkpoint during muscle development. In particular, fetuses lacking Nfix are characterized by a slow-twitch musculature, typical of the embryonic period, while embryos overexpressing Nfix switch to a more mature fetal-like phenotype[17]. In addition, we recently observed that, postnatally, Nfix is crucial for the maintenance of the correct timing of skeletal muscle regeneration upon injury[18].

Here, we show that silencing Nfix in both α-sarcoglycan (Sgca null)- and dystrophin (mdx)-deficient dystrophic mice strikingly protects from the degenerative process by promoting a more oxidative musculature and by slowing down muscle regeneration, in contrast to previous attempts that aimed to promote regeneration. These data are supportive of a new role for Nfix in the progression of MD and suggest Nfix as a novel target to treat this severe disease. More in general, we provide proof of principle for an innovative therapeutic approach based on the idea that slowing down the degeneration–regeneration cycles, instead of increasing regeneration, delays the progression of the pathology.

## Results

**Absence of Nfix improves dystrophic signs of Sgca null mice.** To verify whether the muscle phenotype observed in the Nfix null mouse[17, 18] would be beneficial in a dystrophic context, we generated dystrophic mice lacking Nfix. The Sgca null mouse model[26] was chosen for our analysis because of its very severe phenotype, resembling the human pathology. Muscle histology was analyzed at different weeks of age, to monitor the progression of the disease. At 3 weeks, the first signs of aberrant muscle structure were already present in Sgca null mice (Fig. 1a), characterized by few regenerative fibers (Supplementary Fig. 1A) and presence of inflammatory infiltrates, necrotic areas, and varying fiber calibre. On the contrary, Sgca null:Nfix null mice were characterized by a compact structure with less interstitial space between myofibers and absence of large degenerative areas. At 5 weeks, Sgca null mice showed increased central nucleation (Fig. 1b and Supplementary Fig. 1B), a typical sign of ongoing regeneration, and presence of degenerated and inflammatory areas. This was particularly evident in the diaphragm, one of the most affected muscles in the pathology[27, 28]. At the same age, Sgca null:Nfix null skeletal muscles still appeared less damaged, with reduced central nucleation and few inflammatory areas (Fig. 1b and Supplementary Fig. 1B). At 8 weeks, inflammation, degeneration, and central nucleation were predominant in Sgca null mice, while still evidently reduced in Sgca null:Nfix null mice (Fig. 2a and Supplementary Fig. 1C). At 12 weeks, endomysial fibrosis was diffuse in Sgca null mice, together with a marked fiber atrophy and further exacerbation of all the other histological signs (Fig. 2b). As observed in the previous time points, even at 12 weeks, muscle morphology was improved in the absence of Nfix (Fig. 2b and Supplementary Fig. 1D). The improvement of the histopathological signs of the disease as percentage of centrally nucleated fibers has been quantified at 3, 5, 8, and 12 weeks, revealing a statistically significant amelioration in the Sgca null:Nfix null mice with respect to control Sgca null mice, thus confirming our hypothesis that the delayed regenerative phenotype of Nfix null mice[18] is beneficial in a dystrophic context and is one of the mechanisms of the observed muscular improvement (Supplementary Fig. 1A–D). Moreover, as shown in Supplementary Fig. 1E, F, at 8 weeks, the cross-sectional area (CSA) of the myofibers, which has a non-homogeneous distribution in Sgca null mice, is completely rescued in dystrophic mice lacking Nfix, which appear similar to wild type (WT). Notably, the analysis of other limb muscles at 8 weeks clearly confirmed that the striking histological amelioration of Sgca null:Nfix null mice is a common feature for all muscles (Supplementary Fig. 2A, B). Most importantly, this morphological improvement persisted over time in the Sgca null:Nfix null mice, at least up to 6 months, when in Sgca null animals, particularly in the diaphragm, muscle necrosis is massive, and fat and fibrotic tissue replaced, almost completely, the damaged musculature (Fig. 2c).

Overall, this analysis showed that absence of Nfix in Sgca null mice leads to a striking morphological improvement that persists over time up to at least 6 months.

**Sgca null:Nfix null mice show morpho-functional rescue.** To further confirm and quantify the histological amelioration observed in Sgca null:Nfix null mice, we measured other important hallmarks characterizing dystrophic muscles at 8 weeks. Precisely, in order to evaluate sarcolemmal integrity, we

systemically injected Evan's blue dye in *Sgca* null and *Sgca* null: *Nfix* null mice. As shown in Fig. 3a–c, we observed a statistically significant decrease in the percentage of Evan's blue dye (EBD) positive fibers in *Sgca* null:*Nfix* null mice if compared to dystrophic *Sgca* null mice, reflecting a higher sarcolemmal integrity in mice lacking Nfix. Moreover, collagen I deposit areas were quantified by immunofluorescence, demonstrating a significant reduction in *Sgca* null:*Nfix* null mice (Fig. 3d and Supplementary Fig. 2C). These histological ameliorations were also accompanied by a reduction of the inflammatory parameters, which were measured through an ELISA assay to detect MIP-2 concentration in muscle (Fig. 3e). In addition, we performed immuno-fluorescence analysis for F4/80 (a macrophage marker) and quantified the number of F4/80+ macrophages per muscle area. As shown in Fig. 3f, the number of macrophages is reduced in double-mutant *Sgca* null:Nfix null mice, with respect to *Sgca* null mice. Notably, this difference is statistically significant starting from week 8, when macrophage infiltration begins to be massive in *Sgca* null mice.

Most importantly, as a consequence of this morphological amelioration, we observed the rescue of the functional ability of double-mutant mice, measuring muscle performance with a treadmill test. Graphs in Fig. 4a, b and Supplementary Movie 1 evidence that dystrophic mice lacking Nfix are characterized by a better performance, almost comparable to the one of WT mice. Interestingly, the best muscle performance was observed in *Nfix* null animals, which run with a higher resistance to fatigue even with respect to WT mice.

**Delayed regeneration and oxidative musculature improve MDs**. The results showed so far clearly demonstrate that the amelioration of the dystrophic phenotype observed in absence of Nfix is, in part, due to a delay in muscle regeneration. We have already demonstrated that lack of Nfix leads to a delay of muscle regeneration upon acute cardiotoxin (CTX) injury[18]. We further investigated this concept in dystrophic *Sgca* null:*Nfix* null mice by looking at developmental MyHC expression relative to central nucleation at different time points, in both *Sgca* null and *Sgca* null: *Nfix* null mice. As shown in the graph in Fig. 4c, we confirmed that regeneration is delayed in *Sgca* null:*Nfix* null mice, whose muscles start to massively regenerate later with respect to control *Sgca* null mice (see difference between time point 3 and 5 weeks). This analysis, together with the quantification of the centrally nucleated fibers (Supplementary Fig. 1A–D) and the measurement of the myofiber CSA (Supplementary Fig. 1E, F), provide direct evidence that *Sgca* null:*Nfix* null mice have a delayed regeneration with respect to *Sgca* null mice.

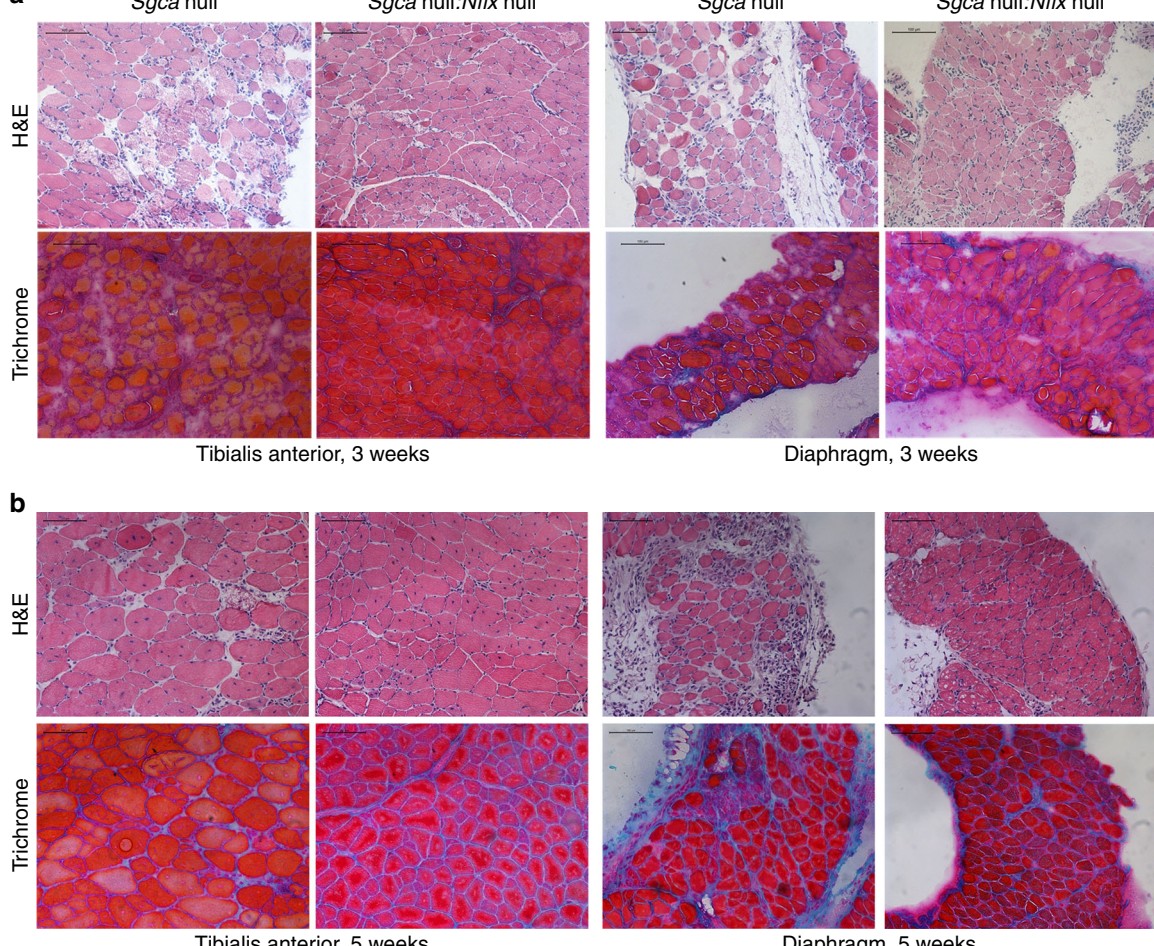

**Fig. 1** Lack of Nfix improves signs of muscular dystrophy. **a** Hematoxylin and eosin (H&E) and Milligan's trichrome staining of tibialis anterior (left) and diaphragm (right) muscles at 3 weeks of age; N = 12 *Sgca* null and 6 *Sgca* null:*Nfix* null mice. Scale bar 100 μm. **b** Hematoxylin and eosin (H&E) and Milligan's trichrome staining of tibialis anterior (left) and diaphragm (right) muscles at 5 weeks of age; N = 5 *Sgca* null and 5 *Sgca* null:*Nfix* null mice. Scale bar 100 μm. See also Supplementary Fig. 1 for the analysis of central nucleation in tibialis anterior and diaphragm muscles at 3 and 5 weeks

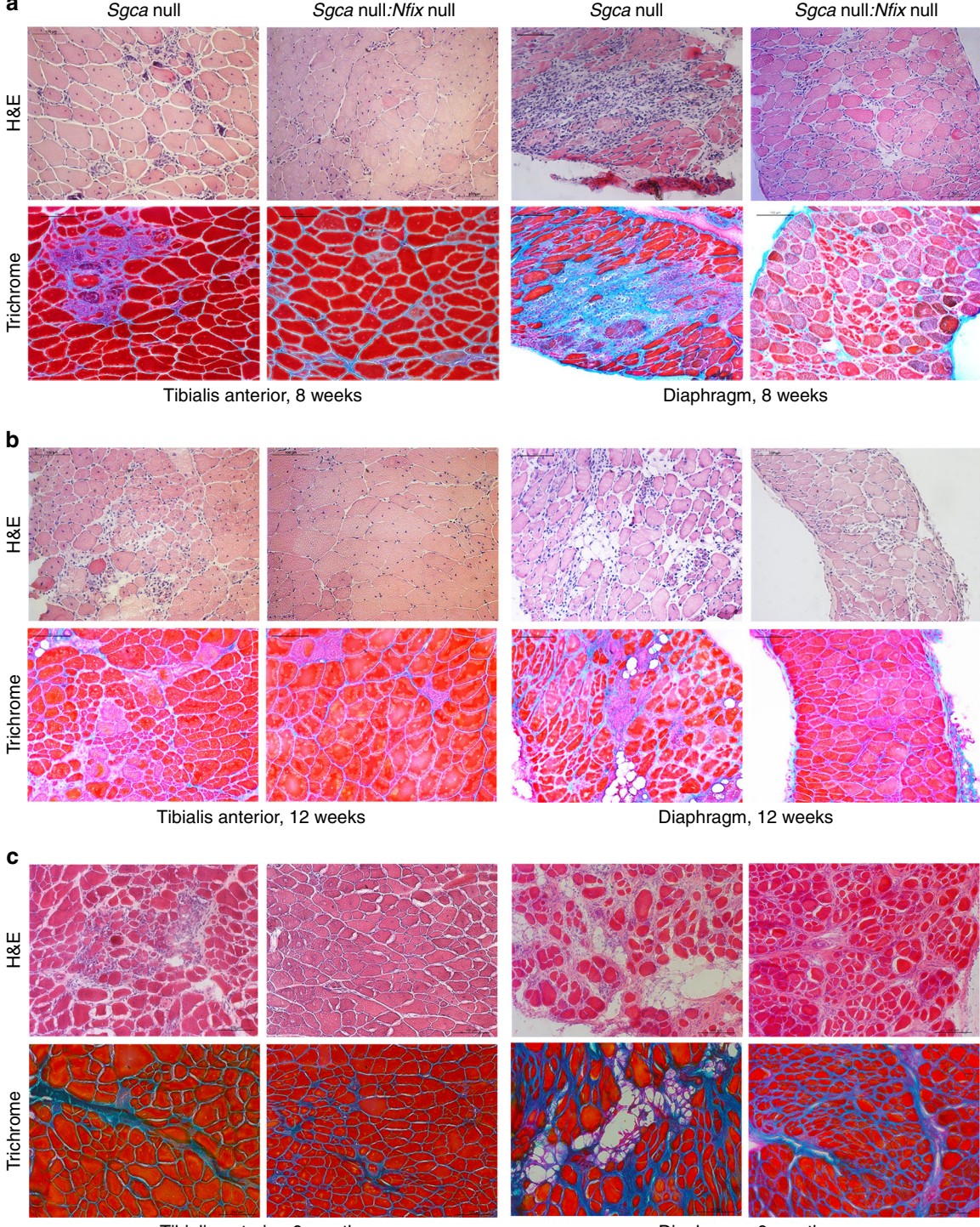

**Fig. 2** Muscular dystrophy amelioration in absence of Nfix persists up to 6 months. **a** Hematoxylin and eosin (H&E) and Milligan's trichrome staining of tibialis anterior (left) and diaphragm (right) muscles at 8 weeks of age; N = 23 *Sgca* null and 12 *Sgca* null:*Nfix* null mice. Scale bar 100 μm. **b** Hematoxylin and eosin (H&E) and Milligan's trichrome staining of tibialis anterior (left) and diaphragm (right) muscles at 12 weeks of age; N = 6 *Sgca* null and 4 *Sgca* null:*Nfix* null mice. Scale bar 100 μm. **c** Hematoxylin and eosin (H&E) and Milligan's trichrome staining of tibialis anterior (left) and diaphragm (right) muscles at 6 months of age; N = 4 *Sgca* null and 4 *Sgca* null:*Nfix* null mice. Scale bar 100 μm. See also Supplementary Fig. 1 for the analysis of central nucleation in tibialis anterior and diaphragm muscles at 8 and 12 weeks and for the analysis of CSA at 8 weeks, Supplementary Fig. 2 for full muscle reconstructions, collagen I quantification, histology of gastrocnemius, quadriceps, soleus, and EDL muscles, and PCR analysis of oxidative fiber genes at 8 weeks, and Supplementary Fig. 3 for the analysis of utrophin and myostatin levels at 8 weeks

To verify whether the improvements observed in dystrophic mice lacking *Nfix* could be also attributed to a more oxidative phenotype, we performed a succinate dehydrogenase (SDH) staining, which highlights in blue the presence of oxidative fibers. Strikingly, dystrophic *Sgca* null mice lacking Nfix displayed a more oxidative phenotype at 3 weeks (Fig. 4d), consistent with the higher resistance to fatigue observed by treadmill test (Fig. 4a, b, Supplementary Movie 1). Additionally, we performed a real-time PCR analysis of typical markers of the oxidative phenotype (SDHA, SDHB, and Cox5)[29–31] in different muscles at 3 weeks. As shown in the graphs in Supplementary Fig. 2D–F, all markers were significantly upregulated in *Sgca* null:*Nfix* null mice, in both slow-twitch and fast-twitch muscles, thus further demonstrating a general switch toward an oxidative metabolism in the *Sgca* null: *Nfix* null mice, regardless the muscle fiber type.

A previous work showed that switch toward slow-twitch fibers upon modulation of $Ca^{2+}$/calmodulin signaling leads to utrophin upregulation in *mdx* mice, thus ameliorating the phenotype[32]. We therefore looked at utrophin expression in single- and double-dystrophic mice, without observing significant differences (Supplementary Fig. 3A). These data exclude utrophin upregulation as a possible mechanism.

In line with this, previous literature claimed that myostatin blockade exerts positive effects on the dystrophic phenotype[10, 33, 34] and, interestingly, we recently reported that during acute injury Nfix is able to modulate myostatin expression during satellite cell differentiation[18]. We therefore verified myostatin expression in *Sgca* null:*Nfix* null mice. As shown in Supplementary Fig. 3B, C, both myostatin messenger RNA expression (Supplementary Fig. 3C) and serum levels

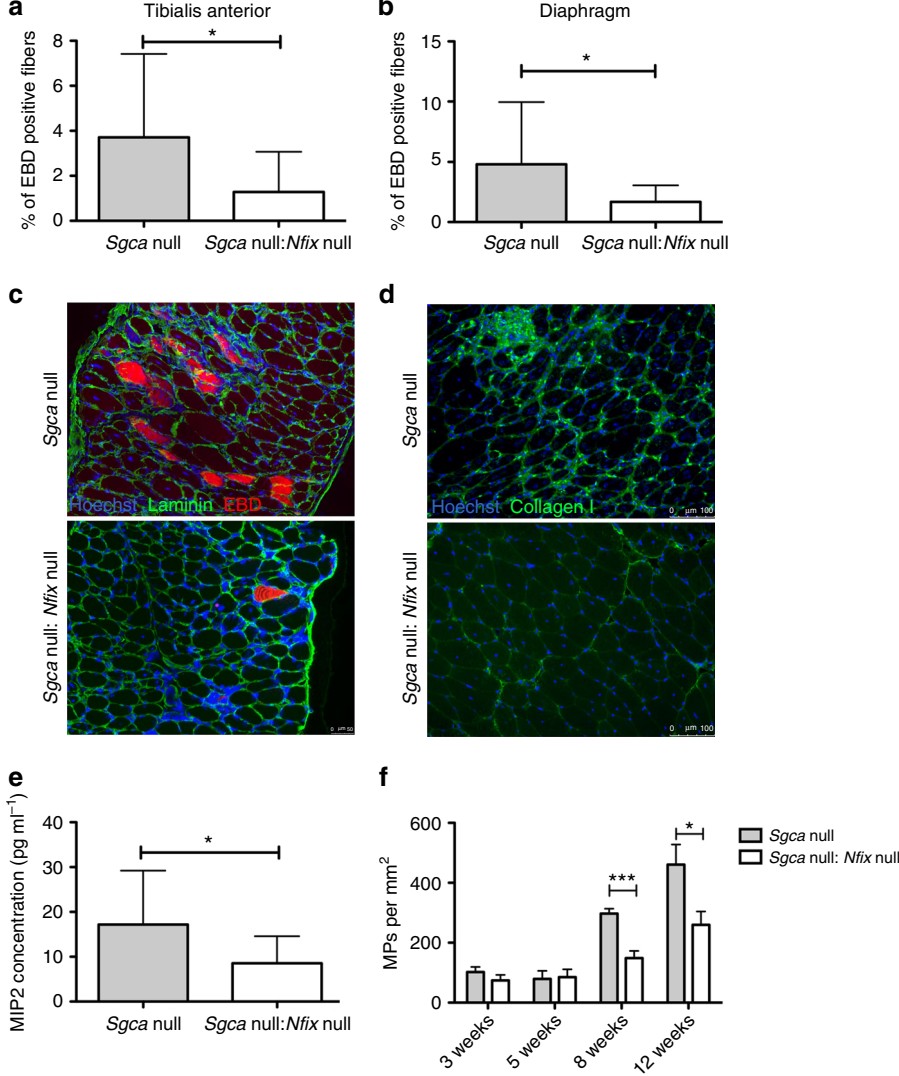

**Fig. 3** Pathological parameters are rescued in dystrophic mice lacking Nfix. **a** Percentage of EBD positive myofibers in tibialis anterior muscles at 8 weeks; $N = 19$ *Sgca* null and 9 *Sgca* null:*Nfix* null mice; mean ± SD; *t* test, *$P < 0.05$. **b** Percentage of EBD positive myofibers in diaphragms at 8 weeks; $N = 19$ *Sgca* null and 11 *Sgca* null:*Nfix* null mice; mean ± SD; *t* test, *$P < 0.05$. **c** Immunofluorescence for laminin (green) and EBD (red) on diaphragms at 8 weeks. Hoechst (blue) stains nuclei. Scale bar 50 μm. **d** Immunofluorescence showing collagen I (green) deposits in tibialis anterior sections. Scale bar 100 μm. **e** MIP2 ELISA assay on gastrocnemius muscles at 8 weeks; $N = 18$ *Sgca* null and 7 *Sgca* null:*Nfix* null mice; mean ± SD; *t* test, *$P < 0.05$. **f** Quantification of the immunofluorescence staining for F4/80, marker of macrophages (MPs), on tibialis anterior muscle sections at different time points. $N = 5$ *Sgca* null and 5 *Sgca* null:*Nfix* null at 3 weeks, $N = 3$ *Sgca* null and 3 *Sgca* null:*Nfix* null at 5 weeks, $N = 8$ *Sgca* null and 8 *Sgca* null:*Nfix* null at 8 weeks, and $N = 12$ *Sgca* null and 7 *Sgca* null:*Nfix* null at 12 weeks. Mean ± SEM; *t* test, *$P < 0.05$, ***$P < 0.001$

(Supplementary Fig. 3B) were similar in all mice analyzed. This result is in keeping with our previous observation that Nfix is able to modulate myostatin expression specifically during the regeneration phase[18]. This implies that the improvement observed in *Sgca* null:*Nfix* null mice is not mediated by a downregulation of myostatin.

**Overexpression of Nfix exacerbates the dystrophic disease.** Based on the positive results that we obtained targeting Nfix in dystrophic mice, we decided to further investigate the link between *Nfix* expression and the progression of the pathology. We therefore generated a dystrophic mouse model overexpressing Nfix downstream a skeletal muscle promoter, taking advantage of Tg:*Mlc1f-Nfix2* mice[17]. Histological analysis of *Sgca* null:*Mlc1f-Nfix2* mice revealed a markedly worsened phenotype with respect to *Sgca* null mice, both at 5 (Fig. 5a) and 8 weeks (Fig. 5b). This was evident in terms of presence of necrotic, fibrotic, and

inflammatory areas (Fig. 5a, b), and was accompanied by an increase of central nucleation (Fig. 5c) and a less homogeneous distribution of myofiber CSAs, with the co-presence in the same muscles of small and hypertrophic fibers (Fig. 5d and Supplementary Fig. 4D). Interestingly, while observing muscle histology of *Sgca* null:*Mlc1f-Nfix2* mice, we encountered a strong phenotypic variability, with impressive exacerbation of the phenotype in some cases, and only a milder exacerbated phenotype in others. We reasoned that this could be explained by the transgenic construct used to generate Tg:*Mlc1f-Nfix2* mice. In fact, in these mice, overexpression of Nfix is guided by a BAC (bacterial artificial chromosome), which carries the *Mlc1f-Nfix2* transgene[17]. Since transgene expression in this case may depend on both BAC copy number and transgene expression efficiency, we decided to classify Nfix overexpressing mice based on their level of transgene overexpression. We evaluated Nfix expression in skeletal muscles of *Sgca* null and *Sgca* null:*Mlc1f-Nfix2* mice, and classified each mouse in one of three subclasses, based on Nfix overexpression

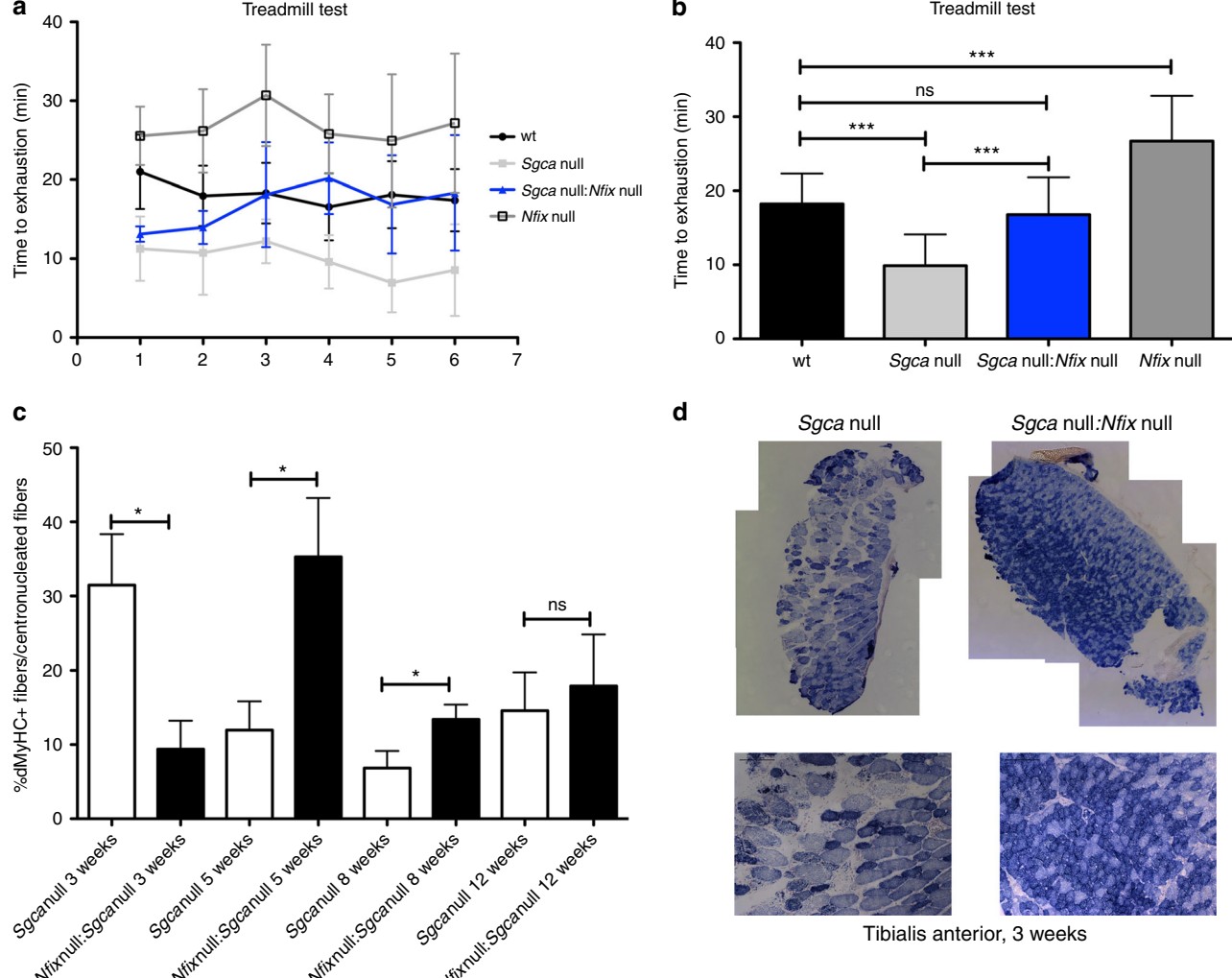

**Fig. 4** *Sgca* null:*Nfix* null mice show improved functionality, an oxidative phenotype, and delayed regeneration. **a, b** Treadmill test on WT, *Sgca* null, *Sgca* null:*Nfix* null, and *Nfix* null mice. **a** represents the performance profile over time, while **b** shows the totality of the measurements; N = 6 measurements per 7 WT, 4 *Sgca* null, 3 *Sgca* null:*Nfix* null, and 4 *Nfix* null mice. Mean ± SD; one-way ANOVA with Bonferroni post-test, ***P < 0.001. ns, non significant. See also Supplementary Movie 1 for an example of a typical run. **c** Percentage of developmental myosin heavy chain positive fibers out of the centrally nucleated fibers at different time points. N = 8 *Sgca* null and 8 *Sgca* null:*Nfix* null mice at 3 weeks, N = 11 *Sgca* null and 11 *Sgca* null:*Nfix* null mice at 5 weeks, N = 11 *Sgca* null and 11 *Sgca* null:*Nfix* null mice at 8 weeks, and N = 10 *Sgca* null and 10 *Sgca* null:*Nfix* null mice at 12 weeks; mean ± SEM; *t* test, *P < 0.05, *P < 0.001. **d** Entire tibialis anterior muscle section reconstructions and higher magnifications showing SDH staining at 3 weeks of age; N = 13 *Sgca* null and 6 *Sgca* null:*Nfix* null mice. Scale bar 100 μm

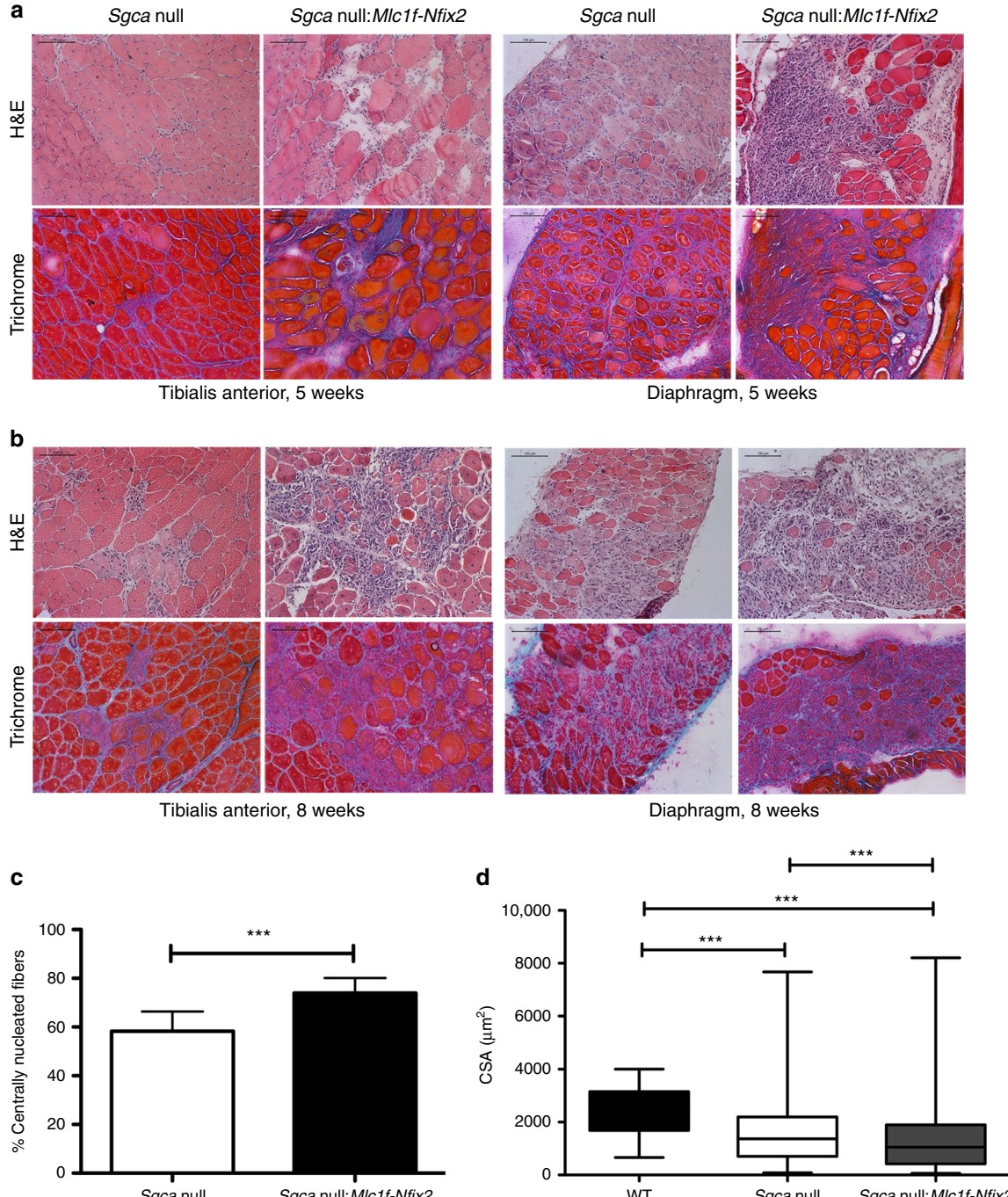

**Fig. 5** Nfix overexpression in skeletal muscle exacerbates the dystrophic phenotype. **a** Hematoxylin and eosin (H&E) and Milligan's trichrome staining of tibialis anterior (left) and diaphragm (right) muscles at 5 weeks of age; $N = 4$ *Sgca* null and 9 *Sgca* null:*Mlc1f-Nfix2* mice. Scale bar 100 μm. **b** Hematoxylin and eosin (H&E) and Milligan's trichrome staining of tibialis anterior (left) and diaphragm (right) muscles at 8 weeks of age; $N = 6$ *Sgca* null and 30 *Sgca* null:*Mlc1f-Nfix2* mice. Scale bar 100 μm. **c** Percentage of centrally nucleated myofibers in tibialis anterior muscles at 8 weeks of age; $N = 11$ *Sgca* null and 18 *Sgca* null:*Mlc1f-Nfix2* mice. Mean ± SD; *t* test, ***$P < 0.001$. **d** Myofiber cross-sectional area distribution at 8 weeks of age; $N = 186$ fibers for WT, 908 for *Sgca* null, and 1026 for *Sgca* null:*Mlc1f-Nfix2* mice. Mean ± whiskers from min to max; one-way ANOVA with Bonferroni post-test, ***$P < 0.001$. See also Supplementary Fig. 4 for PCR and WB analysis of Nfix expression in *Sgca* null:*Mlc1f-Nfix2* mice, as well as for central nucleation and CSA measurements

with respect to *Sgca* null mice, whose expression was considered as 1. As shown in Supplementary Fig. 4A, B, a high variability was observed among *Sgca* null:*Mlc1f-Nfix2* mice, in terms of transgene expression. In some mice, Nfix expression was only slightly increased with respect to *Sgca* null mice (class 1–4), while others had an intermediate (class 4–10) or very high expression of the

transgene (>10-fold expression with respect to *Sgca* null mice). Interestingly, mice with a very high overexpression of Nfix (belonging to the >10 class) showed a more marked exacerbation of the phenotype, which we quantified in terms of central nucleation (Supplementary Fig. 4C). In fact, mice belonging to the >10 class were characterized by a higher central nucleation

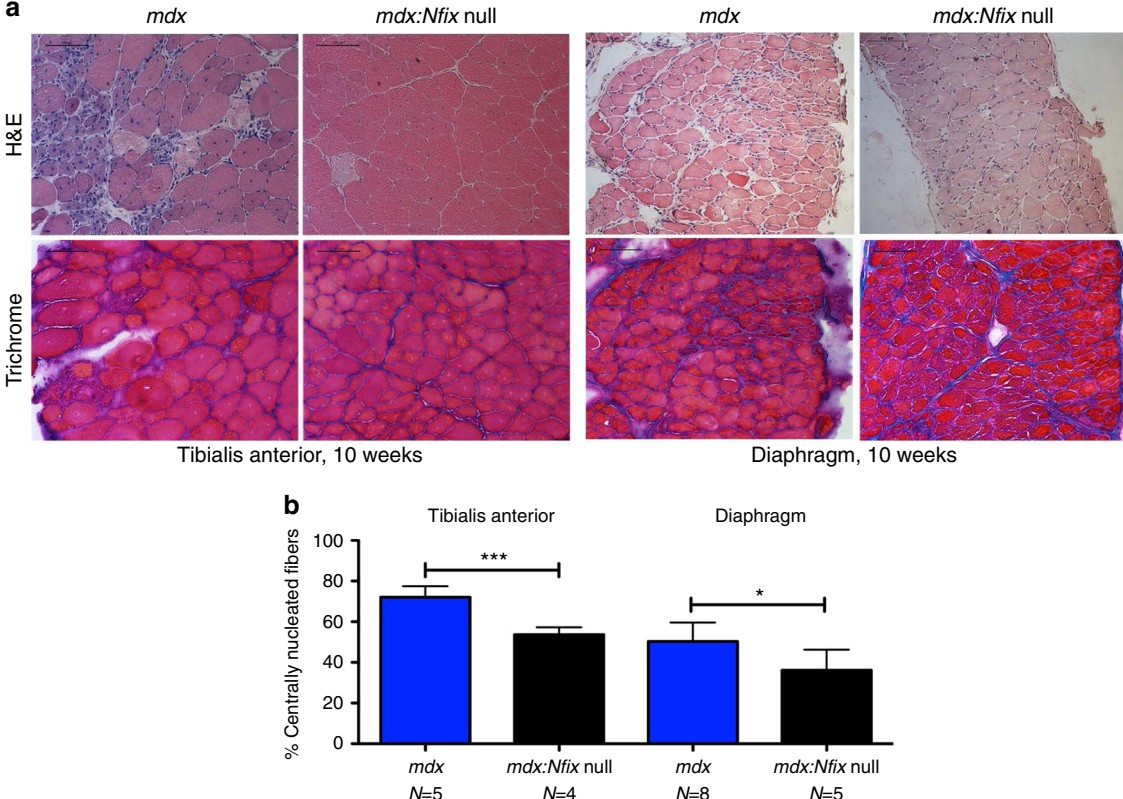

**Fig. 6** Targeting Nfix ameliorates muscular dystrophy also in *mdx* mice. **a** Hematoxylin and eosin (H&E) and Milligan's trichrome staining of tibialis anterior (left) and diaphragm (right) muscles at 10 weeks of age; N = 5 *mdx* and 4 *mdx:Nfix* null mice. Scale bar 100 μm. **b** Percentage of centrally nucleated myofibers in tibialis anterior (TA) and diaphragm muscles at 10 weeks of age; N = 5 *mdx* and 4 *mdx:Nfix* null mice for TA. N = 8 *mdx* and 5 *mdx: Nfix* null mice for diaphragm. Mean ± SD, *t* test, *P < 0.05; ***P < 0.001. See also Supplementary Fig. 3 for the analysis of utrophin and myostatin levels in *mdx* mice

with respect not only to *Sgca* null mice, but also with respect to *Sgca* null:*Mlc1f-Nfix2* mice belonging to class 1–4. Histological stainings shown in Fig. 5 refer to the highest class of Nfix expression, chosen as the most representative.

**Lack of Nfix also protects the *mdx* dystrophic muscles.** In light of the striking amelioration that we observed in *Sgca* null mice, we verified whether our approach could be applied to other MDs. To this aim, we deleted *Nfix* in *mdx* mice, the mouse model for DMD, the most common form of muscular dystrophy in humans[35].

As shown in Fig. 6a, *mdx:Nfix* null mice showed a milder phenotype compared to *mdx* controls. The improvement was evident in terms of muscle structure and presence of necrotic, fibrotic, and inflammatory areas, in both tibialis anterior and diaphragm muscles (Fig. 6a). To understand whether the phenotypic amelioration was mediated by a delayed regeneration, similarly to what observed for *Sgca* null:*Nfix* null mice, we measured central nucleation in tibialis anterior and diaphragm muscles of *mdx* and *mdx:Nfix* null mice. As expected, we observed a significant decrease in the percentage of centrally nucleated fibers in *mdx:Nfix* null mice, compared to *mdx* (Fig. 6b).

Overall, the data shown demonstrate that lack of *Nfix* leads to a rescue of the histopathological signs of the dystrophic disease, with reduced central nucleation and a better muscle histology, regardless of the genetic defect.

**Silencing Nfix in adult *Sgca* null mice rescues MD signs.** The results shown so far have clearly demonstrated that genetic ablation of Nfix in dystrophic animals causes a significant improvement of the disease, thus protecting from muscle degeneration. In light of a future translational approach, we wondered whether absence of Nfix might lead to the same effects observed even when the dystrophic disease already occurred, which represents what would normally be feasible in patients.

To this aim, we electroporated the tibialis anterior of 5-week-old *Sgca* null mice with control (scramble) or sh*Nfix* plasmids[17, 18]. Muscles were analyzed 1 and 2 days after electroporation to verify Nfix silencing (Fig. 7a), and 2 weeks later to verify its effect on muscle (Fig. 7b, c). We observed a striking rescue of the dystrophic muscle morphology in terms of reduced infiltrates, centrally nucleated myofibers, and CSA distribution in *Sgca* null muscles silenced for Nfix if compared with the contralateral muscles where the main hallmarks of muscular dystrophy already appeared (Fig. 7b–f). Consistently, this phenotypical rescue is also due to a more oxidative musculature, as observed by the SDH staining (Fig. 7g). Moreover, this result is accompanied by a robust reduction of the collagen I deposits (Fig. 7h).

**Discussion**
MDs are severe inherited muscle disorders that still lack an effective therapy. The typical feature of the pathology is the succession of cycles of degenerations followed by attempts to

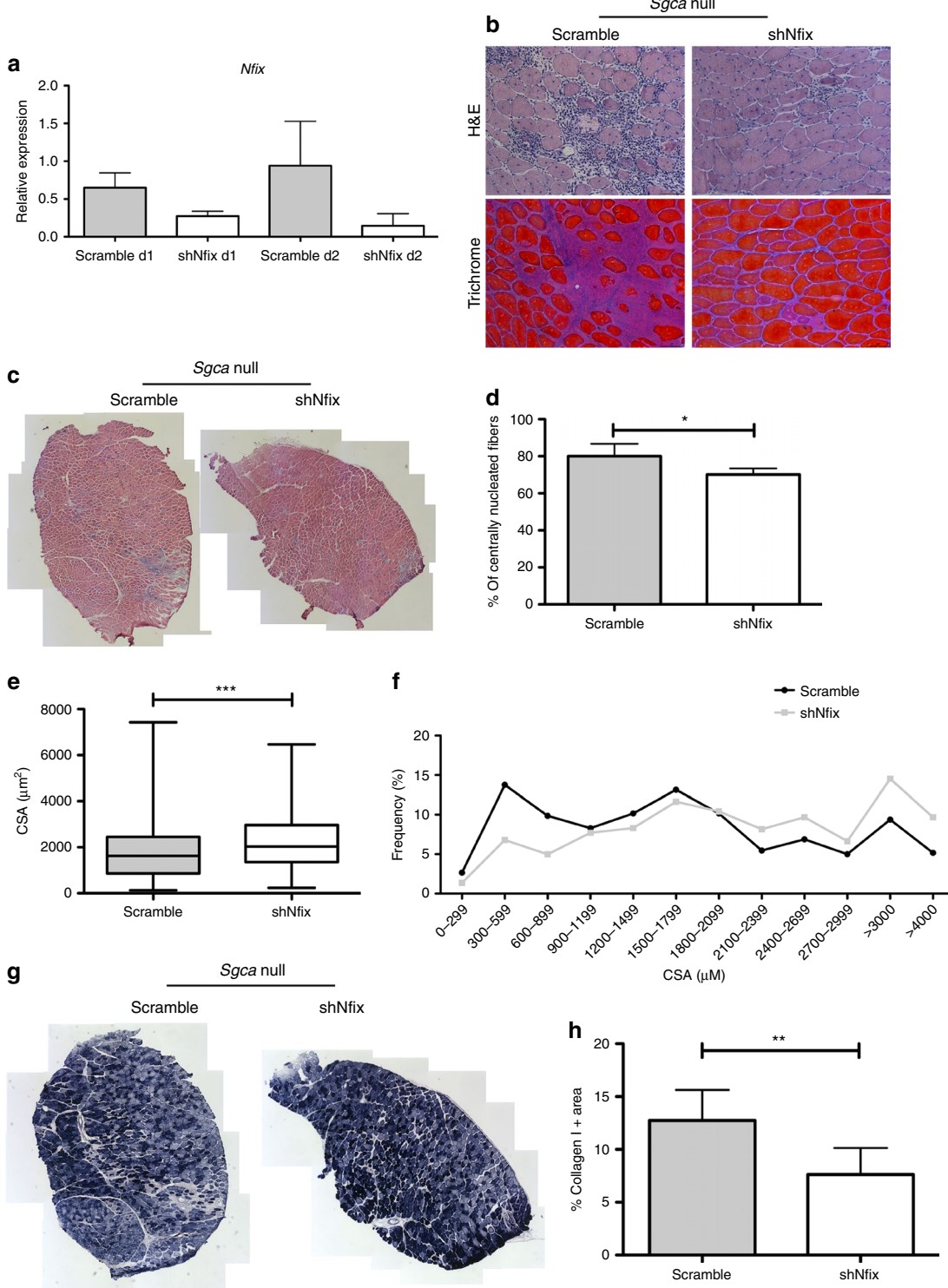

**Fig. 7** Silencing Nfix in adult *Sgca* null mice morphologically rescues the dystrophic pathology. **a** Real-time PCR analysis of *Nfix* expression on *Sgca* null muscles electroporated with scramble or shNfix plasmids after 1 or 2 days (d) from electroporation. $N = 2$ for each group. Mean ± SD. **b** Hematoxylin and eosin (H&E) and Milligan's trichrome staining of tibialis anterior muscles from *Sgca* null mice electroporated with scramble or shNfix plasmids; $N = 9$ mice. Scale bar 75 μm. **c** Entire tibialis anterior muscle section reconstructions of *Sgca* null mice electroporated with scramble or shNfix plasmids; $N = 9$ mice. **d** Percentage of centrally nucleated myofibers in tibialis anterior muscle sections from *Sgca* null mice electroporated with scramble or shNfix plasmids. $N = 5$ mice. Mean ± SD; *t* test, *$P < 0.05$. **e, f** Myofiber cross-sectional area distribution in *Sgca* null mice electroporated with scramble or shNfix plasmids; $N = 639$ fibers for scramble and 661 for shNfix. Mean ± whiskers from min to max; *t* test, ***$P < 0.001$. **g** SDH staining of *Sgca* null tibialis anterior muscles electroporated with scramble or shNfix vectors; $N = 4$ mice. **h** Quantification of collagen I positive areas in *Sgca* null tibialis anterior muscles electroporated with scramble ($N = 9$) or shNfix ($N = 9$) plasmids. Mean ± SD; *t* test, **$P < 0.01$

regenerate the damage, which not only fail to successfully repair the muscle, but also foster the degenerative process.

Here, we show that targeting the transcription factor Nfix in dystrophic mice induces a protective effect from the progression of the pathology with striking improvements of both morphological and functional muscle properties. This is achieved through a mechanism acting at different levels: Nfix is able not only to slow down the regenerative burst characterizing the disease[18], but also to promote a shift to a more oxidative phenotype, that is known to confer protection from oxidative stress-induced damage[13, 15].

We report a robust improvement in dystrophic mice lacking Nfix, which was evident in terms of general muscle structure, reduced inflammatory areas, central nucleation, and fibrotic deposition. These features were persistent up to 6 months. The morphological amelioration was also accompanied by reduced membrane damage and inflammation, as demonstrated by EBD, MIP2 assays, and quantification of the number of macrophages in muscles. Most importantly, mice lacking Nfix showed increased resistance to fatigue and better muscle performance with respect to controls, in keeping with our observation that these mice are characterized by a more oxidative phenotype. Conversely, we show that the dystrophic phenotype is exacerbated when Nfix is overexpressed specifically in skeletal muscle, and that the exacerbation correlates with levels of Nfix overexpression. Very interestingly, we observed that both *Sgca* null and *mdx* dystrophic mice lacking *Nfix* are characterized by comparable rescue of the histopathological signs, regardless of their genetic defect.

These results came from dystrophic animals that genetically lack Nfix since the very early stages of the pathology. Notably, we demonstrated that silencing of Nfix strongly rescues the pathological signs even when the disease has already occurred, providing the basis for a future translational approach for MD patients.

To our knowledge, this is the first study resulting in such a significant amelioration of the dystrophic pathology as a consequence, at least in part, of a switch toward an oxidative phenotype. In fact, previous studies targeting calcineurin expression were conducted to try to promote slow MyHC positive fibers, although only in the less severe (with respect to *Sgca* null mice) dystrophic *mdx* mouse model[32, 36]. These studies proposed upregulation of utrophin as compensatory mechanism, while in our hands, utrophin is not upregulated. This supports the concept that targeting Nfix protects from MD acting on different mechanisms.

Importantly, we are proposing here that delaying regeneration induces a beneficial effect as a consequence of the delay of the degeneration–regeneration circuit. This is in evident contrast with previous studies that, in the past years, tried to recover muscle function by promoting regeneration[10, 11]. However, none of these approaches has been successful when translated to humans[12]. Moreover, these studies were based on blockade of myostatin, which in our hands is not differentially regulated in dystrophic mice. Notably, our findings are in line with other recent observations demonstrating that blockade of myostatin in other dystrophic animal models worsens the disease[37, 38].

Therefore, we are proposing that Nfix exerts its functions through a double mechanism, involving both a switch to an oxidative phenotype and a delay of the regenerative process. We cannot absolutely exclude other mechanisms involved, since Nfix is also expressed by other cell populations such as macrophages (Saclier et al., unpublished results). Nevertheless, the exacerbated phenotype observed in the double *Sgca* null:*Mlc1f-Nfix2* mice, where Nfix is overexpressed specifically in muscle cells, clearly supports the evidence that a slow regenerating and a slow twitching dystrophic musculature is at the basis of the amelioration/rescue observed. After all, even the soleus (a typical slow

muscle) is itself affected in the dystrophic animals as also human patients who mainly display a slow musculature, suggesting that an oxidative phenotype is not sufficient to protect the musculature from the degeneration.

Our results, beyond the approach used, provide clear and robust evidence for a wider and ground-breaking concept: forcing regeneration in a structurally impaired dystrophic muscle actually leads to an increase of the oxidative stress and an acceleration of degeneration cycles, which exacerbates the phenotype. The data presented here demonstrate that the successful strategy is the opposite. In this sense, any other approach resulting in both slower regeneration and an oxidative musculature should be preferred in a dystrophic context.

In particular, the strategy of silencing Nfix could be applied in combination with adeno-associated virus (AAV)-mediated gene correction[39–41]. Nfix silencing could in fact be used to stabilize muscle degeneration/regeneration, allowing a longer expression of the AAV transgene, which on the contrary would be rapidly lost after treatment as a consequence of the continuous regeneration of the infected fibers. This would result in an increased efficiency of the treatment. Alternatively, a "druggable" way to selectively inhibit Nfix expression or its function might be developed in the future, upon the identification of the signaling regulating Nfix and/or the Nfix protein structure.

In the long term, the development of drugs targeting Nfix would represent a better, valid strategy. All these aspects give to our study a strong translational potential.

## Methods

**Mouse models.** *Sgca* null, *Nfix* null, *mdx*, and Tg:Mlc1f-*Nfix2* mice were previously described[17, 26, 35, 42]. As described in refs [18, 42], *Nfix* null mice were fed with a special transgenic mice dough diet (Bioserv). To avoid any possible effect due to the diet composition, the diet was given to all WT, *Sgca* null, and *mdx* mice used as control mice for single *Nfix* null and double *Sgca* null:*Nfix* null or *mdx*:*Nfix* null mice. Both male and female mice were used. Mice were kept in pathogen-free conditions and all procedures were conformed to Italian law (D. Lgs no. 2014/26, implementation of the 2010/63/UE) and approved by the University of Milan Animal Welfare Body and by the Italian Ministry of Health.

**In vivo electroporation.** In vivo electroporation was performed on 5-week-old *Sgca* null mice, as also described in Rossi et al.[18] Animals were anesthetized and 40 μg of control (scramble) or shNfix plasmids resuspended in 0.9% salt solution[17] were injected in a total volume of 40 μl in tibialis anterior muscles. Muscles were then immediately electroporated using a pulse generator (ECM 830, BTX) equipped with 5 μm needle electrodes to generate 100 V pulses, with a fixed duration of 20 ms and an interval of 200 ms between the pulses.

**Hematoxylin and eosin and Milligan's trichrome.** Hematoxylin and eosin staining was performed on 7-μm-thick cryosections fixed with 4% paraformaldehyde for 10 min at 4 °C. The staining was performed according to standard protocols. For Milligan's trichrome staining, sections were fixed for 1 h with Bouin's fixative (Sigma-Aldrich) and rinsed for 1 h under running water. Sections were then rapidly dehydrated to 95% EtOH in graded ethanol solutions, successively passed in 3% potassium dichromate (Sigma-Aldrich) for 5 min, rapidly washed in distilled water, stained with 0.1% acid fuchsin (Sigma-Aldrich) for 30 s, washed again in distilled water, passed in 1% phosphomolybdic acid (Sigma-Aldrich) for 3 min, stained with Orange G (2% in 1% phosphomolybdic acid; Sigma-Aldrich) for 5 min, rinsed in distilled water, passed in 1% acetic acid (VWR) for 2 min, stained with 1% Fast Green for 5 min (Sigma-Aldrich), passed in 1% acetic acid for 3 min, rapidly dehydrated to 100% EtOH, and passed in xylene before mounting with Eukitt (Bio-Optica).

**SDH staining.** SDH staining was performed on 3-week-old mice. For SDH staining, freshly cut 7-μm-thick cryosections were used. Sections were incubated in SDH incubating solution (one tablet of nitrobluetetrazolium dissolved in 0.1 M sodium succinate–0.1 M phosphate buffer, pH 7.4, all from Sigma-Aldrich) for 1 h at 37 °C, rinsed in distilled water, rapidly passed in 30%, 60%, 30% acetone (VWR), and rinsed again in distilled water. Sections were then rapidly dehydrated in graded EtOH solutions, cleared in xylene, and mounted with Eukitt mounting medium.

**ELISA assays.** Measurement of MIP-2 concentration was performed on gastrocnemius muscle protein extracts. Protein extracts were obtained homogenizing

minced muscle tissues in tissue lysis buffer (150 mM Tris-HCl, pH 7.5; 1 mM EDTA, 1% Triton, 150 mM NaCl (all from Sigma-Aldrich)) for 30 s, followed by lysis on ice for 30 min and by centrifugation at 10,000 r.p.m. at 4 °C to pellet cell debris. Supernatant was quantified using DC Protein Assay (Bio-Rad) and 500 µg of proteins were loaded to perform a MIP-2 ELISA assay (R&D Systems), following manufacturer's instructions. Myostatin levels in serum were measured using Quantikine ELISA GDF-8/Myostatin Immunoassay (R&D Systems), following manufacturer's instructions. Blood samples were allowed to clot O/N at 4 °C before centrifuging for 20 min at 2000×g. Serum samples were separated and maintained at −20 °C before loading on the ELISA plate.

**Evan's blue dye uptake measurement**. Evan's blue dye (Sigma-Aldrich) solution (5 mg ml$^{-1}$) was injected intraperitoneally in 8-week-old mice 24 h before sacrifice (10 µl per g of mice). Positivity for Evan's blue dye was revealed through its autofluorescence, fixing sections with acetone (VWR) for 10 min at −20 °C, permeabilizing them in 1% BSA (Sigma-Aldrich)−0.2% Triton X-100 (Sigma-Aldrich) for 30 min, and incubating them O/N with rabbit anti-laminin antibody (1:300, Sigma-Aldrich) to reveal myofiber outlines. The day after, sections were washed, incubated with a goat anti-rabbit 488 secondary antibody (1:250, Jackson Lab), together with Hoechst (1:250, Sigma-Aldrich), washed again, and finally mounted with a fluorescence mounting medium (Dako). Measurement of the percentage of Evan's blue dye uptake was performed counting the number of Evan's blue dye positive fibers on total muscle section reconstructions, using Image J software. Diaphragm and TA muscles were counted as the most representative.

**Treadmill test**. For Treadmill test functional assay, 4-week-old mice were exercised three times, once a week, before recording of their performances. Treadmill test was therefore performed starting from 7-week-old mice, once a week for 6 weeks. The test was conducted on a treadmill (Bioseb) with a 10% incline, starting from a speed of 6 cm s$^{-1}$ and increasing it by 2 cm s$^{-1}$ every 2 min. For each test, the time to exhaustion of each mouse was measured.

**RNA extraction, reverse transcription PCR, and real-time PCR**. Real-time PCR was performed starting from RNA extracts obtained from muscle tissues homogenized and extracted in Trizol Reagent (Invitrogen) following manufacturer's instructions. RNA (1 µg) was retro-transcribed to complementary DNA (cDNA) using iScript cDNA Synthesis Kit (Bio-Rad), and 5 µl of diluted (1:5) cDNA was used for each sample. Gene expression was quantified by comparative CT method, using GAPDH as a reference. Primers used are: *Nfix* fwd CTGGCTTACTTTGTCCACACTC; *Nfix* rev CCAGCTCTGTCACATTCCAGAC; *GAPDH* fwd AGGTCGGTGTGAACGGATTTG; *GAPDH* rev TGTAGACCATG TAGTTGAGGTCA; *utrophin* fwd AAGATGGGAGAAAGCTCTTGGA; *utrophin* rev TCGGTTGACATTGTTTAAGGCA; myostatin fwd AAGATGACGATTATG ACGCTACC; *myostatin* rev CCGCTTGCATTAGAAAGTCAGA; *SDHA* fwd AGAGATGTTGTGTCTCGATCCAT; *SDHA* rev CTGCAGGTAGACGTGAT CTTTCT; *SDHB* fwd AGCAAAGTCTCCAAAATCTACCC; *SDHB* rev TCAATG GATTTGTATTGTGCGTA; *Cox5A* fwd TTGCGTAAAGGGATGAATACACT; *Cox5A* rev TTTGTCCTTAACAACCTCCAAGA.

**Western blot**. Western blot was performed on protein extracts from muscles homogenized in tissue buffer and processed as described for MIP-2 ELISA assay. Total protein extracts of 30 µg were loaded for each sample. Images were acquired using Chemidoc ImageLab software (Bio-Rad). The following antibodies and dilutions were used: rabbit anti-Nfix (1:5000, Geneka Biotechnology), mouse anti-Vinculin (1:2500, Clone VIN-11-5, Sigma-Aldrich), and IgG-HRP secondary antibodies (1:10,000, Bio-Rad).

**Immunofluorescence**. Immunofluorescence was performed on 7 µm cryosections. Slices were fixed for 10 min at 4 °C with 4% PFA (apart from staining for F4/80 and dMHC that do not require fixation), washed twice in PBS, and permeabilized with a solution containing 1% BSA and 0.2% Triton X-100 in PBS, for 30 min at room temperature. After a blocking for 30 min with 10% donkey or goat serum, slices were incubated O/N with primary antibodies in PBS-1.5% donkey or goat serum. The day after, two washes in PBS-1% BSA-0.2% Triton X-100 were performed, and samples were incubated for 45 min at room temperature with secondary antibodies (donkey anti goat 488, goat anti rabbit 594, goat anti mouse 488, donkey anti rat 488, and donkey anti rabbit 594 (1:250, Jackson Lab)) and Hoechst (Sigma-Aldrich). Excess of antibody was washed twice in PBS-0.2% Triton X-100 before mounting with fluorescence mounting medium (Dako). The following primary antibodies and dilutions were used: goat anti-collagen I (1:200, Southern Biotech), rat anti-F4/80 (1:400, Clone CI-A3-1, Novusbio), rabbit anti-laminin (1:300, Sigma-Aldrich), and mouse anti-dMHC, which detects the Myh3 isoform (1:40, Clone MONX10806, Monosan).

**Image acquisition**. Images were acquired with an inverted microscope (Leica-DMI6000B) equipped with Leica DFC365FX and DFC400 cameras and ×10 and ×20 magnification objectives. The Leica Application Suite software was used for acquisition, while Photoshop was used to generate merged images.

**Measurement of central nucleation, CSA, and collagen I**. Measurement of central nucleation and myofiber CSA was performed on tibialis anterior muscle sections of mice at 8 weeks (at 10 weeks for *mdx* and *mdx:Nfix* null mice), using Image J software. Collagen I quantification was performed using a Macro in ImageJ to identify and quantify collagen I positive areas.

**Statistics**. All data shown in graph are expressed as mean ± SD, apart from graphs showing CSA distributions, which are expressed as mean ± whiskers from min to max. Statistical analysis between two columns was performed using two-tailed unpaired Student's *t* test, whereas data containing more than two experimental groups were analyzed with one-way analysis of variance followed by Bonferroni's test. *$P < 0.05$; **$P < 0.01$; ***$P < 0.001$; confidence intervals 95%; alpha level 0.05.

**Data availability**. The authors declare that all data supporting the findings of this study are available within the paper and its Supplementary Information files.

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

## Acknowledgements

We thank T. Lopez Royo and R. Bacchetta for technical assistance and G. Cossu for helpful discussion. We are also grateful to R. Gronostajski for the kind exchange of information and animal models. This work was funded by the European Community, ERC StG2011 (RegeneratioNfix 280611).

## Author contributions

G.R. designed and performed all the experiments with the assistance of S.A., M.B., M.S., and A.I. for in vivo and histological analysis, and C.B., S.M., and V.T. for molecular biology. G.M. supervised the work and wrote the paper together with G.R.

## Additional information

**Competing interests:** The authors declare no competing financial interests.

