## [Peer Review file · Nature Communications]

Reviewers' comments:

Reviewer #1 (Remarks to the Author):

This manuscript aims at proposing a novel concept in the therapeutic approaches for muscular dystrophies. Much interestingly, such concept does not rely on enhancing muscle regeneration, but points at slowing regeneration as a novel perspective.

The manuscript is very well written, easy to read and coherent the presentation and discussion of results. In the introduction, the authors overview the current state of the art on the therapeutic strategies for muscular dystrophies and they clearly state where their work can be placed in this respect and in which way their findings could set a step forward in the field. The points the authors want to address appear clear, and I appreciated the choice of the authors to validate their hypothesis in two different animal models, namely *Sgca* null and *mdx* mice. However, although the whole body of this manuscript provides undeniable evidence of the solidness of the authors' hypothesis, the underlying mechanism(s) mostly relies on deductive basis, as such mechanisms are discussed but not demonstrated through experimental evidence in the current version of the manuscript. I am strongly convinced about the high potential of this work, but I believe that the authors should provide more experimental evidence at a mechanistic level.

SPECIFIC COMMENTS

1. The authors claim that the histological ameliorations seen in *Sgca*^{-/-}::*Nfix*^{-/-} mice compared to *Sgca*^{-/-} mice "confirm our hypothesis that the delayed regenerative phenotype of *Nfix* null mice is beneficial in a dystrophic context and is one of the mechanisms of the observed muscular improvement". While the benefit on the dystrophic phenotype appears unquestionable from both the histological and functional analysis reported in Figures 1-3, further experiments should be carried out to make the entire work more solid and throw light on the underlying mechanism(s). The authors properly refer to their previous work describing the phenotype of *Nfix* null mice, but nothing related to regeneration is taken into account here, with the exception of centrally nucleated fibers. In their previous work, the authors showed that lack of *Nfix* expression in satellite cells leads to reduced cross sectional area and increased expression of slow MyHC, which is negatively regulated by *Nfix*. In addition, they showed implications for delayed satellite cell activation/differentiation and even an impairment of muscle regeneration after injury, by monitoring myogenin and developmental MyHC expression. While I believe that the concept of delaying regeneration in muscular dystrophy is highly innovative and potentially promising, I would be excited to see signs of delayed regeneration in the dystrophic models studied here, but unfortunately in the present version of the manuscript this is just a deduction. A broadly used tool for proving the concept of delayed muscle regeneration is performing muscle injury and investigating the effects after 7 days. The regeneration aspect has even more relevance when handling a dystrophic model as the *mdx* mouse, which is characterized by sequential degeneration/regeneration cycles, with the earliest signs of regeneration at six weeks of age (see for example Turk et al., BMC Genomics 2005). This aspect does not seem to have been taken into full consideration neither in the results nor in the discussion, although such findings could strengthen the "delayed regeneration" hypothesis.

2. The results described in Fig. 1B point at reduced central nucleation at 5 weeks in *Sgca*^{-/-}::*Nfix*^{-/-} mice when compared to *Sgca*^{-/-} animals. Results referring to Fig. 2A and Fig. S1A do not report quantitative data for centrally nucleated fibers in 8-week muscles. This is reported in Fig. 3A for tibialis anterior, but not for diaphragms. Although not strongly supported by the provided images, on the contrary this phenotype appear still evident at 8 weeks and 12 weeks of age. When showing the recovery of muscle phenotype addressing central nucleation, the authors should provide quantification of the mentioned histological features for all the time points. Quantification of central nucleation is also needed for diaphragms of *mdx* mice.

3. When referring to signs of inflammation, immunofluorescence staining for immunological

markers on sections are expected to be provided.

4. I suggest to modify the graph representing the distribution of the cross-sectional areas with another one showing the percentage of myofibers in different area ranges (i.e. range of $300\mu\text{m}$: $0 < x < 299$; $300 < x < 599$, ...). In this way the authors could better support their data.

5. The percentage of centrally nucleated fibers of Sgca null mice in Fig. 4C is different from that displayed in Fig. 3A. When comparing the percentage of centrally nucleated fibers between Sgca^{-/-} mice shown in Fig. 3A and Sgca^{-/-}:MLC1f-Nfix2 mice, the statistics may strongly change. What does N mean in authors' analysis? N=10 In Fig. 3A for Sgca^{-/-}, whereas N=5 in Fig. 4C; could this mean that by broadening the analysis of Fig. 4C for Sgca^{-/-} mice this percentage would become more similar to the one of Fig. 3A?

6. SDH staining and treadmill analysis, shown only for Sgca null mice, should be also provided for mdx mice, in order to strengthen the authors' conclusions.

MINOR POINTS

1. Please correct "deficient dystrophic mice strikingly protects form the degenerative process" on page 1.

2. I suggest moving panel 5C, concerning the SDH staining in Sgca^{-/-}:Nfix^{-/-} mice to another figure, since the entire Figure 5 is related to mdx.

4. I suggest putting side by side the quantification of the histopathological dystrophic signs with the relative histological images.

Reviewer #2 (Remarks to the Author):

The manuscript entitled « Silencing Nfix rescues muscular dystrophy by delaying muscle regeneration » and submitted by Rossi et al describes a very interesting phenomenon : the amelioration of a dystrophic phenotype in two distinct mouse models by silencing the Nfix gene. In accordance, they also describe a worsening of these phenotypes if Nfix is overexpressed. The experiments are carefully conducted and convincing. The paper is of wide interest since it proposes that enhancing regeneration in a degenerative disease can be detrimental. This is in opposite to several efforts which have been conducted recently to improve/enhance regeneration to ameliorate dystrophies, efforts which have been largely unsuccessful. This paper shows convincing data that explain why and proposes a new paradigm to improve the dystrophic phenotype, based on this new concept that forcing regeneration does not help. It should be noted that this is more or less in line with what has been observed in dystrophic patients. As such, the paper would certainly be of interest for a wide community working on gene or cell therapy for muscular dystrophies.

There are few additions that would even more improve the manuscript :

- The level of inflammation in the different animals, particularly in the experiments comparing Nfix null to dystrophic control, is measured by the detection of MIP-2 by ELISA. A detection of the inflammatory infiltrate by specific antibodies on sections would have been more informative and more illustrative.

- Same remark concerning the delay in regeneration. The authors measure central nucleation at various time points, but if central nucleation is indeed the result of regeneration, nuclei stay then central for a long time. It would be much more accurate to detect the expression of embryonic or neonatal myosin heavy chain by immunohistochemistry at these various time points. This would give a more accurate description of the dynamics of regeneration (eg by counting the number of positive newly formed fibres). The expression of these two myosins, for which specific antibodies

exist, is transient during regeneration, as opposed to centronucleation.

- The discussion is somewhat deceiving and could be largely improved, eg by discussing why ameliorating, even transiently, muscle phenotype could be important for the development of therapeutic strategies. One could quote for instance the loss of AAV genome just after treatment, due to the degeneration/regeneration of infected fibres. If the muscle a better stabilized by slowing down the process, this could be enough for the AAV to have its transgene expressed at a high enough level to be protective. This could also be applied to many therapeutic strategies where a transient amelioration could increase the efficiency of these strategies. This should be extensively discussed/reviewed in the discussion to improve the interest of the presented results.

Provided these more or less minor comments are taken into account, the paper is clear, the experiments are well described (and rather simple, including to reproduce) and the paper could be published in Nature Communication.

NB: Some improvement in english could be done, although the overall quality is rather good.

Reviewer #3 (Remarks to the Author):

The authors have examined whether making a dystrophic muscle slower and more oxidative might be of therapeutic benefit. To do this, they have knocked out the transcription factor Nfix in mouse models of Duchenne muscular dystrophy and limb girdle muscular dystrophy. They provide evidence to support their hypothesis that silencing Nfix improves the pathology, whereas its overexpression worsens the pathology.

This will be of interest to those interested in possible therapeutic avenues for muscular dystrophies.

The manuscript is not written very clearly and there are several points that need amendment or clarification before the manuscript would be suitable for publication (detailed below).

Introduction

Page 3

Clarify what is meant by "diffused"?

Is this the function of corticosteroids in muscular dystrophy only reduction of inflammation? Are they used in all muscular dystrophies, or just in DMD?

What is meant by "good muscle quality"? And why are these patients restricted from clinical trials?

Damage, not damages (throughout)

Explain – is the damage caused by the satellite cells?

Page 4

Slow twitch, not twitching

Most human skeletal muscles are slow, in contrast to most mouse muscles, which are fast. This should be discussed, as it impacts on the extrapolation of data from mice to humans.

The last sentence on this page needs to be clarified.

Results

Page 5

Discuss whether the pathology in the mouse model of sarcoglycan deficiency is similar to that in human sarcoglycan deficient patients.

Explain "diffused".

Explain "tissue disorganization".

"mice revealed" – does this mean the muscles of these mice had...?

How were the "degenerative areas" quantified?

Central nucleation is a measure of regeneration that has occurred sometime in the past (many references in the literature) – this is not a measure of ongoing regeneration. This should be made clear.

Give references that the diaphragm in the mouse is one of the muscles most affected by lack of dystrophin.

Clarify what is meant by "preserved and homogeneous".

"Evidently" reduced – is this statistically significant? If so, this should be stated.

Was there a significant reduction of centrally-nucleated fibers in limb muscles, or was this only in the diaphragm?

"Strongly" – is this significantly?

What other limb muscles were analysed? It would be particularly important to include the soleus (a slow mouse muscle).

Page 6

"Persists over time" – this should be qualified, as the mice were only examined up to 12 weeks of age. A longer timescale would be needed to demonstrate persistence (6 months, 1 year, over 2 years).

What is the evidence that regeneration is delayed, rather than being reduced? Unless evidence can be supplied for this point, this statement should be removed throughout the manuscript.

Clarify what is meant by "single dystrophic".

Clarify what is meant by "preserved musculature"

Page 7

Did the mice show resistance to eccentric exercise?

Page 8

The most common form of muscular dystrophy in humans.

What is meant by "a healthy phenotype"? Non-pathological?

Delayed regeneration – see above.

Were the analyses of the mice performed by an operator that was blinded to the mouse phenotype? Whether this was so or not should be stated.

Page 9

"Signs of the dystrophic disease" should be clarified.

Which muscles had a more oxidative phenotype at 3 weeks?

Was there a change in the amount of myostatin protein in the muscles?

What is meant by the "differentiation phase"? Does this refer to muscle development? If it means regeneration, this should be made clear.

Suggests, not suggest

Page 10

Clarify "pass through"

Delete "delay" unless there are data to support this.

"Already appeared" – does this mean that there was muscle pathology present before the intervention?

Was the reduction in fibrosis statistically significant?

Discussion

Page 11

What is meant by a "resolved" therapy?

Discuss the mouse soleus muscle –does this muscle have protection from oxidative damage?

"Persistent over time" should be deleted or qualified, as this was not a long-term investigation.

Page 12

"Wide, striking amelioration" should be qualified.

Clarify what model mdx is less severe than.

What is meant by "pushing on regeneration"?

Clarify the sentence beginning "Our results, beyond the..."

Clarify the sentence beginning "The original idea..."

Methods

Page 14

Why was a dough diet used?

Are there any differences (in the literature) between male and female mice of the different models?

Page 16

10 ml/g cannot be correct.

Page 17

Sections, not slices.

Figures

There is obvious ice crystal artefact in many of the sections (e.g. figure 2, top panels A and B, figure 4 top left panel B, Figure 5, top right panel A) –panels with ice crystal artifact should be replaced with images of muscles that do not have this artefact.

There seems to be an artefact in figure 6 C (knife cutting artefact?) –

Such images are not suitable for publication.

Figure 2 – “over time” should be qualified.

Figures 3 and 4 – CSA data are better shown as graphs with size bins.

Figure 3

What was the % of EDB+ fibers in the TAs?

Give data for collagen quantification.

Figure 4 D – why were so few WT fibers quantified?

We would like to thank the Reviewers for their constructive comments. Listed below are point by point replies to the comments raised by the Reviewers. The new parts in the manuscript now appear in blue.

Reviewers' comments:

Reviewer #1 (Remarks to the Author):

This manuscript aims at proposing a novel concept in the therapeutic approaches for muscular dystrophies. Much interestingly, such concept does not rely on enhancing muscle regeneration, but points at slowing regeneration as a novel perspective.

The manuscript is very well written, easy to read and coherent the presentation and discussion of results. In the introduction, the authors overview the current state of the art on the therapeutic strategies for muscular dystrophies and they clearly state where their work can be placed in this respect and in which way their findings could set a step forward in the field. The points the authors want to address appear clear, and I appreciated the choice of the authors to validate their hypothesis in two different animal models, namely *Sgca* null and *mdx* mice. However, although the whole body of this manuscript provides undeniable evidence of the solidness of the authors' hypothesis, the underlying mechanism(s) mostly relies on deductive basis, as such mechanisms are discussed but not demonstrated through experimental evidence in the current version of the manuscript. I am strongly convinced about the high potential of this work, but I believe that the authors should provide more experimental evidence at a mechanistic level.

SPECIFIC COMMENTS

1. The authors claim that the histological ameliorations seen in *Sgca*^{-/-};*Nfix*^{-/-} mice compared to *Sgca*^{-/-} mice “confirm our hypothesis that the delayed regenerative phenotype of *Nfix* null mice is beneficial in a dystrophic context and is one of the mechanisms of the observed muscular improvement”. While the benefit on the dystrophic phenotype appears unquestionable from both the histological and functional analysis reported in Figures 1-3, further experiments should be carried out to make the entire work more solid and throw light on the underlying mechanism(s).

The authors properly refer to their previous work describing the phenotype of *Nfix* null mice, but nothing related to regeneration is taken into account here, with the exception of centrally nucleated fibers. In their previous work, the authors showed that lack of *Nfix* expression in satellite cells leads to reduced cross sectional area and increased expression of slow MyHC, which is negatively regulated by *Nfix*. In addition, they showed implications for delayed satellite cell activation/differentiation and even an impairment of muscle regeneration after injury, by monitoring myogenin and developmental MyHC expression. While I believe that the concept of delaying regeneration in muscular dystrophy is highly innovative and potentially promising, I would be excited to see signs of delayed regeneration in the dystrophic models studied here, but unfortunately in the present version of the manuscript this is just a deduction. A broadly used tool for proving the concept of delayed muscle regeneration is performing muscle injury and investigating the effects after 7 days. The regeneration aspect has even more relevance when handling a dystrophic model as the *mdx* mouse, which is characterized by sequential degeneration/regeneration cycles, with the earliest signs of regeneration at six weeks of age (see for example Turk et al., BMC Genomics 2005). This aspect does not seem to have been taken into full consideration neither in the results nor in the discussion, although such findings could strengthen the “delayed regeneration” hypothesis.

As also suggested by other Reviewers, we decided to further investigate the concept of delayed regeneration in dystrophic *Sgca*/*Nfix* null mice. To do that, we looked at developmental MyHC expression relative to central nucleation at different time points, in both *Sgca* null and *Nfix* null mice. As shown in the graph below (now also shown in Fig.4), we confirmed that regeneration is delayed in *Sgca*/*Nfix* null mice, which start to massively regenerate later with respect to control *Sgca* null mice (see difference between time point 3 weeks and 5 weeks). This analysis, together with the quantification of the centrally nucleated fibers, that we now implemented for both *Tibialis anterior* and Diaphragm muscles at different time points from 3 to 12 weeks, and the measurement of the myofiber cross sectional area (now shown in Fig.S1), provide direct and non-deductive evidence that *Sgca* null/*Nfix* null mice have a delayed regeneration with respect to *Sgca* null mice.

Percentage of developmental Myosin Heavy Chain positive fibers out of the centrally nucleated fibers at different time points. N=8 *Sgca* null and 8 *Sgca* null:*Nfix* null mice at 3 weeks, N=11 *Sgca* null and 11 *Sgca* null:*Nfix* null mice at 5 weeks, N=11 *Sgca* null and 11 *Sgca* null:*Nfix* null mice at 8 weeks, N=10 *Sgca* null and 10 *Sgca* null:*Nfix* null mice at 12 weeks. *P<0,05; ns, non significant.

2. The results described in Fig. 1B point at reduced central nucleation at 5 weeks in *Sgca*^{-/-}:*Nfix*^{-/-} mice when compared to *Sgca*^{-/-} animals. Results referring to Fig. 2A and Fig. S1A do not report quantitative data for centrally nucleated fibers in 8-week muscles. This is reported in Fig. 3A for tibialis anterior, but not for diaphragms. Although not strongly supported by the provided images, on the contrary this phenotype appear still evident at 8 weeks and 12 weeks of age. When showing the recovery of muscle phenotype addressing central nucleation, the authors should provide quantification of the mentioned histological features for all the time points. Quantification of central nucleation is also needed for diaphragms of mdx mice.

As requested by the Reviewer, we quantified central nucleation at all time point analyzed, for both *Tibialis anterior* and Diaphragm muscles. As shown in the graphs below, now in Figures S1 and Figure 6 in the MS, central nucleation is always statistically significantly reduced in absence of *Nfix* in both muscles for both Muscular Dystrophy models (MDX and *Sgca* null), at all the time point analysed.

Percentage of centrally nucleated myofibers in *Tibialis anterior* and Diaphragm muscles at different time points and in both Muscular Dystrophy models; mean± SD. *P<0,05; **P<0,01; ***P<0,001;

3. When referring to signs of inflammation, immunofluorescence staining for immunological markers on sections are expected to be provided.

We agree and we thank the Reviewer for this useful comment. We performed immunofluorescence analysis for F4/80 (a macrophage marker) and quantified the number of F4/80+ macrophages per muscle fiber. As shown in the graph below (now shown in Fig. 3 in the MS), the number of macrophages are reduced in double mutant *Sgca* null:*Nfix* null mice, with respect to *Sgca* null mice. Notably, this difference is statistically significant starting from week 8, when macrophage infiltration begins to be massive in *Sgca* null mice.

Quantification of the immunofluorescence staining for F4/80 on *Tibialis anterior* muscle sections at different time points. N=6 *Sgca* null and 5 *Sgca* null:*Nfix* null at 3 weeks, N=5 *Sgca* null and 5 *Sgca* null:*Nfix* null at 5 weeks, N=8 *Sgca* null and 8 *Sgca* null:*Nfix* null at 8 weeks, N=12 *Sgca* null and 7 *Sgca* null:*Nfix* null at 12 weeks. mean± SEM. *P<0,05; **P<0,01; ns: non significant.

4. I suggest to modify the graph representing the distribution of the cross-sectional areas with another one showing the percentage of myofibers in different area ranges (i.e. range of 300µm: 0<x<299; 300<x<599, ...). In this way the authors could better support their data.

We thank the Reviewer for this useful comment. We modified, as suggested, the cross-sectional area graphs in Fig.3B, 4D, 6E. Moreover, we increased the N for WT myofibers in Fig.4D, as suggested by Reviewer 3. We believe that both graphical representation may be useful to the reader to visualize and interpret the data, therefore we decided to maintain both graph types and to include the new ones in Fig.7, Fig.S1 and Fig.S4.

Modified graph from Fig.3B

Modified graph from Fig.4D

Modified graph from Fig.6E

5. The percentage of centrally nucleated fibers of Sgca null mice in Fig. 4C is different from that displayed in Fig. 3A. When comparing the percentage of centrally nucleated fibers between Sgca^{-/-} mice shown in Fig. 3A and Sgca^{-/-}:MLC1f-Nfix2 mice, the statistics may strongly change. What does N mean in authors' analysis? N=10 In Fig. 3A for Sgca^{-/-}, whereas N=5 in Fig. 4C; could this mean that by broadening the analysis of Fig. 4C for Sgca^{-/-} mice this percentage would become more similar to the one of Fig. 3A?

We thank the Reviewer for comment and we better explained and addressed these results. For both graphs, N is the number of animals. Since, as pointed out by the Reviewer, the number of animals was different in the two experimental groups, we increased the N of Sgca null mice in Fig. 4C to 11, to reach the same number of mice (and even one more) analysed for Fig. 3A. As shown in the graph below, that we also substituted to Fig. 5C, even increasing the number of animals, the value of the percentage of central nucleation does not change, while we have an increase in significance, with $P < 0.0001$. We are not surprised by this difference, that we attribute to the mouse background, which is different for the 2 models. In fact, since littermates or age-matched mice from similar couplings were always compared, Sgca null mice in Fig. 4C (now 5C) are on a FVB background (as are Sgca null:Mlc1f-Nfix2 mice), while Sgca null mice in Fig. 3A (now Fig.S1C) are on a C57BL/6 background (as are Sgca null:Nfix null mice), which may explain this difference. Indeed, it is known that, even in other dystrophic models, the appearance of dystrophic signs is dependent on the mouse background (Coley WD et al., Hum Mol Genet. 2016, Effect of genetic background on the dystrophic phenotype in mdx mice).

Percentage of centrally nucleated myofibers in *Tibialis anterior* (TA) muscles at 8 weeks of age; N=10 *Sgca* null and 18 *Sgca* null:Mlc1f-Nfix2 mice. Mean±SD. ***P<0,001.

6. SDH staining and treadmill analysis, shown only for *Sgca* null mice, should be also provided for *mdx* mice, in order to strengthen the authors' conclusions.

We partially agree with the Reviewer on this observation. We choose the *Sgca* dystrophic mice because of their severe phenotype, which better than the *mdx* mouse model resemble the pathophysiology present in human Duchenne patients. This work does not aim to genetically correct the dystrophic phenotype, but rather it is providing a completely new proof of concept as a strategy to preserve both muscle quality and function. To this aim, the *Sgca* is the best animal model. We just performed few but essential experiments on the *mdx* mice to demonstrate that the morphological amelioration observed in the *Sgca* mice lacking Nfix also occurs in the *mdx* mouse model, therefore regardless the genetic defect. In this sense, it is beyond the scope of the work to repeat all the experiments performed on the *Sgca* mice even in the *mdx* animal. Moreover, the first signs of the dystrophic disease in the *mdx* mice are displayed not before the 6 weeks of age. Therefore, a treadmill test would imply a long-lasting experiment, which will cost time for the revision, and a huge number of animals, in sharp contrast with the 3Rs directive for the use animals for scientific purposes (2010/63/UE), not providing in the end any improvement and/or more information/data to this work. However, we analyzed the oxidative fiber state in *mdx* versus *mdx: Nfix* null mice at 5 weeks old, to address at least one concern raised by the Reviewer. As shown in the Figure below, the increase of SDH, measured by both qRT-PCR and by SDH, in the double *mdx: Nfix* null mice is not statistically significant if compared to the *mdx* mice (although a trend can be appreciated). But it is important to note, that the basic level of SDH in the *mdx* animals is higher itself than the one in the *Sgca* mice, thus maybe also explaining the milder phenotype of the *mdx* animals. Our hypothesis is that to better appreciated a significant difference in SDH between *mdx* and *mdx: Nfix* null mice, we should look at later time points which would imply a longer revision and whose relevance would not improve the quality and the main message of this work.

Upper panel: Real Time PCR showing expression of SDH-A/B in *Sgca* null, *Sgca* null:*Nfix* null, *mdx* and *mdx:Nfix* null mice at 3 weeks. N=5 *Sgca* null, 5 *Sgca* null:*Nfix* null, 12 *mdx* and 12 *mdx:Nfix* null for SDH-A and 6 *Sgca* null, 6 *Sgca* null:*Nfix* null, 12 *mdx* and 8 *mdx:Nfix* null for SDH-B. Mean±SD. Lower panel: SDH staining on *mdx* and *mdx:Nfix* null mice at 3 weeks. *P<0,05; **P<0,01.

MINOR POINTS

1. Please correct “deficient dystrophic mice strikingly protects form the degenerative process” on page 1. We thank the Reviewer and the sentence has been corrected.

2. I suggest moving panel 5C, concerning the SDH staining in *Sgca*^{-/-}:*Nfix*^{-/-} mice to another figure, since the entire Figure 5 is related to *mdx*. We moved the pictures as suggested.

4. I suggest putting side by side the quantification of the histopathological dystrophic signs with the relative histological images.

We tried to build the Figures as suggested by the Reviewer, but the histological pictures resulted too much reduced to well appreciate the histopathological dystrophic signs. Therefore, we decided to include the quantification in a dedicated supplementary Figure (new Fig. S1). We apologize for that, but we firmly believe that the histological pictures deserve the best resolution and visibility, since they are really significant.

Reviewer #2 (Remarks to the Author):

The manuscript entitled « Silencing *Nfix* rescues muscular dystrophy by delaying muscle regeneration » and submitted by Rossi et al describes a very interesting phenomenon : the amelioration of a dystrophic phenotype in two distinct mouse models by silencing the *Nfix* gene. In accordance, they also describe a worsening of these phenotypes if *Nfix* is overexpressed. The experiments are carefully conducted and convincing. The paper is of wide interest since it proposes that enhancing regeneration in a degenerative disease can be detrimental. This is

in opposite to several efforts which have been conducted recently to improve/enhance regeneration to ameliorate dystrophies, efforts which have been largely unsuccessful. This paper shows convincing data that explain why and proposes a new paradigm to improve the dystrophic phenotype, based on this new concept that forcing regeneration does not help. It should be noted that this is more or less in line with what has been observed in

dystrophic patients. As such, the paper would certainly be of interest for a wide community working on gene or cell therapy for muscular dystrophies.

There are few additions that would even more improve the manuscript :

- The level of inflammation in the different animals, particularly in the experiments comparing Nfix null to dystrophic control, is measured by the detection of MIP-2 by ELISA. A detection of the inflammatory infiltrate by specific antibodies on sections would have been more informative and more illustrative.

We agree and we thank the Reviewer for this useful comment. We performed immunofluorescence analysis for F4/80 (a macrophage marker) and quantified the number of F4/80+ macrophages per muscle fiber. As shown in the graph below (now shown in Fig. 3 in the MS), the number of macrophages are reduced in double mutant *Sgca* null:*Nfix* null mice, with respect to *Sgca* null mice. Notably, this difference is statistically significant starting from week 8, when macrophage infiltration begins to be massive in *Sgca* null mice.

Quantification of the immunofluorescence staining for F4/80 on *Tibialis anterior* muscle sections at different time points. N=6 *Sgca* null and 5 *Sgca* null:*Nfix* null at 3 weeks, N=5 *Sgca* null and 5 *Sgca* null:*Nfix* null at 5 weeks, N=8 *Sgca* null and 8 *Sgca* null:*Nfix* null at 8 weeks, N=12 *Sgca* null and 7 *Sgca* null:*Nfix* null at 12 weeks. mean± SEM. *P<0,05; **P<0,01; ns: non significant.

- Same remark concerning the delay in regeneration. The authors measure central nucleation at various time points, but if central nucleation is indeed the result of regeneration, nuclei stay then central for a long time. It would be much more accurate to detect the expression of embryonic or neonatal myosin heavy chain by immunohistochemistry at these various time points. This would give a more accurate description of the dynamics of regeneration (eg by counting the number of positive newly formed fibres). The expression of these two myosins, for which specific antibodies exist, is transient during regeneration, as opposed to centronucleation.

We thank the Reviewer for this useful suggestion that we believe will significantly improve the Manuscript. We performed immunofluorescences for developmental MyHC (dMyHC) and quantified its expression at different time points as the percentage of dMyHC out of the number of centrally nucleated fibers, thus representing the portion of fibers newly regenerating at each time point. From the graph below, now inserted in Fig.4, the differences in muscle regeneration between *Sgca* null and *Sgca* null:*Nfix* null mice are robustly evident. This data are supporting our hypothesis that regeneration is delayed in the absence of Nfix (as also demonstrated for acute regeneration in Rossi et al., Cell Reports 2016). In fact, the regenerative burst occurs at earlier time points in *Sgca* null mice (3 weeks on), while regeneration starts later in *Sgca* null:*Nfix* null mice (5 weeks on).

Percentage of developmental Myosin Heavy Chain positive fibers out of the centrally nucleated fibers at different time points. N=8 *Sgca* null and 8 *Sgca* null:*Nfix* null mice at 3 weeks, N=11 *Sgca* null and 11 *Sgca* null:*Nfix* null mice at 5 weeks, N=11 *Sgca* null and 11 *Sgca* null:*Nfix* null mice at 8 weeks, N=10 *Sgca* null and 10 *Sgca* null:*Nfix* null mice at 12 weeks. *P<0,05; ns: non significant.

- The discussion is somewhat deceiving and could be largely improved, eg by discussing why ameliorating, even transiently, muscle phenotype could be important for the development of therapeutic strategies. One could quote for instance the loss of AAV genome just after treatment, due to the degeneration/regeneration of infected fibres. If the muscle a better stabilized by slowing down the process, this could be enough for the AAV to have its transgene expressed at a high enough level to be protective. This could also be applied to many therapeutic strategies where a transient amelioration could increase the efficiency of these strategies. This should be extensively discussed/reviewed in the discussion to improve the interest of the presented results.

We thank the Reviewer for the useful suggestion. We added a paragraph in the discussion highlighting the importance of *Nfix* silencing in combination with AAV treatments.

Provided these more or less minor comments are taken into account, the paper is clear, the experiments are well described (and rather simple, including to reproduce) and the paper could be published in Nature Communication.

NB: Some improvement in english could be done, although the overall quality is rather good.

Reviewer #3 (Remarks to the Author):

The authors have examined whether making a dystrophic muscle slower and more oxidative might be of therapeutic benefit. To do this, they have knocked out the transcription factor *Nfix* in mouse models of Duchenne muscular dystrophy and limb girdle muscular dystrophy. They provide evidence to support their hypothesis that silencing *Nfix* improves the pathology, whereas its overexpression worsens the pathology.

This will be of interest to those interested in possible therapeutic avenues for muscular dystrophies.

The manuscript is not written very clearly and there are several points that need amendment or clarification before the manuscript would be suitable for publication (detailed below).

Introduction

Page 3

Clarify what is meant by “diffused”? We decided to eliminate the ambiguous term “diffused” and to substitute it with a better explanation.

Is this the function of corticosteroids in muscular dystrophy only reduction of inflammation? Are they used in all muscular dystrophies, or just in DMD? Corticosteroids are prevalently used to reduce inflammation in patients. Actually, their application in different Muscular Dystrophies is still debated.

What is meant by “good muscle quality”? And why are these patients restricted from clinical trials? We more extensively explained this concept in the text.

Damage, not damages (throughout) We checked and corrected it.

Explain – is the damage caused by the satellite cells? The damage is not caused by satellite cells per se. What we wanted to highlight in this sentence was that, being SCs over-stimulated because of the persistent damage, (which they cannot repair since they share the same genetic mutation as the myofiber), their function cannot rescue the dystrophic phenotype. Moreover, as we suggest in this work, the continuous regeneration driven by SCs is in the end not beneficial, since not resolving. To avoid this misinterpretation by the reader, we decided to remove the sentence “contributing in turn to the exacerbation of the pathology” from the text.

Page 4

Slow twitch, not twitching We checked and corrected it throughout the text.

Most human skeletal muscles are slow, in contrast to most mouse muscles, which are fast. This should be discussed, as it impacts on the extrapolation of data from mice to humans. This has been better discussed in the text.

The last sentence on this page needs to be clarified. We clarified the sentence in the text.

Results

Page 5

Discuss whether the pathology in the mouse model of sarcoglycan deficiency is similar to that in human sarcoglycan deficient patients. The sentence in the text was modified. In particular, no reference to a similarity to the DMD patients has been highlighted. In general, in fact, the *Sgca* null mouse model muscle phenotype is more severe than the *mdx* phenotype, therefore being a better model to study human Muscular Dystrophies that usually display strong histological phenotypes (both in DMD and sarcoglycan deficient patients).

Explain “diffused”. Diffused has been substituted with “present”.

Explain “tissue disorganization”. The sentence has been changed.

“mice revealed” – does this mean the muscles of these mice had....? Yes, this was the original meaning. The sentence was changed in the text to avoid confusion.

How were the “degenerative areas” quantified? Degenerative areas were not specifically quantified. Nevertheless, as also evident from histological pictures, no degeneration was present in Hematoxylin and Eosin stained *Sgca* null:*Nfix* null sections at 3 weeks.

Central nucleation is a measure of regeneration that has occurred sometime in the past (many references in the literature) – this is not a measure of ongoing regeneration. This should be made clear.

As suggested by another Reviewer, we added to the Manuscript the quantification of the expression of developmental MyHC (dMyHC), a myosin isoform that is a marker of ongoing regeneration, which supports our hypothesis that the regenerative burst in *Sgca* null:*Nfix* null mice is delayed (now in Fig.4).

Give references that the diaphragm in the mouse is one of the muscles most affected by lack of dystrophin.

The requested references were added in the text.

Clarify what is meant by “preserved and homogeneous”. To better clarify the meaning, we substituted “preserved and homogeneous” with “less damaged”.

“Evidently” reduced – is this statistically significant? If so, this should be stated. Quantifications of all these parameters are provided in the figures and differences are statistically significant as shown in Fig.3E and 3F for inflammation, Fig.3A, 3B and S2C for muscle damage and Fig.S1C for central nucleation

Was there a significant reduction of centrally-nucleated fibers in limb muscles, or was this only in the diaphragm? The measure of centrally nucleation in the first Manuscript version was related to limb muscles (Tibialis anterior). In the revised version of the Manuscript, we included quantification of centrally nucleation also in the diaphragm, for all time points analysed (now in Fig. S1 and 6B).

“Strongly” –is this significantly? We removed strongly from the text due to the impossibility in quantifying a qualitative parameter such as muscle histology.

What other limb muscles were analysed? It would be particularly important to include the soleus (a slow mouse muscle).

We analysed even the soleus as shown in Fig. S2. It is important to consider that the soleus is itself affected in the dystrophic animals, suggesting that an oxidative phenotype is not sufficient to protect the musculature from muscle degeneration. This latter consideration also supports that even a delayed muscle regeneration is required, as also our results demonstrated. We are indeed proposing that Nfix exerts its functions through a double mechanism, involving both a switch to an oxidative phenotype and a delay of the regenerative process.

Page 6

“Persists over time” – this should be qualified, as the mice were only examined up to 12 weeks of age. A longer timescale would be needed to demonstrate persistence (6 months, 1 year, over 2 years).

We agree with the Reviewer on this point. We therefore decided to add to our analysis a longer timescale, up to 6 months. Notably, differences between *Sgca* null mice and *Sgca* null:*Nfix* null mice at this time point are not only still evident, but even more clear, as shown in the Figure below (and now also in Fig. 2)

Haematoxylin and Eosin and Milligan's Trichrome stainings of *Tibialis anterior* and Diaphragm muscles from *Sgca* null and *Sgca* null:*Nfix* null mice at 6 months.

What is the evidence that regeneration is delayed, rather than being reduced? Unless evidence can be supplied for this point, this statement should be removed throughout the manuscript.

As also suggested by another Reviewer, in order to clarify this point we performed immunofluorescence analysis for developmental MyHC (dMyHC), and quantified the number of newly regenerating fibers at each time point.

As shown in the graph below (now added to Fig. 4), the onset of the regenerative burst is delayed from week 3 to week 5 in *Sgca* null:*Nfix* null mice, if compared to *Sgca* null controls. This is also in accordance with what we previously demonstrated in the context of acute regeneration (Rossi et al., Cell Reports 2016).

Percentage of developmental Myosin Heavy Chain positive fibers out of the centrally nucleated fibers at different time points. N=8 *Sgca* null and 8 *Sgca* null:*Nfix* null mice at 3 weeks, N=11 *Sgca* null and 11 *Sgca* null:*Nfix* null mice at 5 weeks, N=11 *Sgca* null and 11 *Sgca* null:*Nfix* null mice at 8 weeks, N=10 *Sgca* null and 10 *Sgca* null:*Nfix* null mice at 12 weeks. *P<0,05; ns: non significant.

Clarify what is meant by “single dystrophic”. Single was originally referred to “Single mutant” mice, carrying a mutation for *Sgca* but not *Nfix*. To avoid misinterpretations, we eliminated single from the text.

Clarify what is meant by “preserved musculature” “Preserved musculature” was substituted with “sarcolemmal integrity”, since the sentence was referred to EBD quantification, a measure of sarcolemmal damage.

Page 7

Did the mice show resistance to eccentric exercise? We did not measure this parameter, and we do not have an answer to this point.

Page 8

The most common form of muscular dystrophy in humans. The sentence was modified as suggested.

What is meant by “a healthy phenotype” ? Non-pathological? Healthier was substituted with milder to avoid confusion.

Delayed regeneration – see above. As mentioned above, in the revised version of our Manuscript, we provided evidence for delayed (instead of reduced) regeneration in *Sgca* null:*Nfix* null mice.

Were the analyses of the mice performed by an operator that was blinded to the mouse phenotype? Whether this was so or not should be stated. Some, but not all, experiments were performed in a blind way. In any case, a blind control was always done internally, showing already processed slices and pictures for qualitative analysis to a blind operator (to double check the phenotype without any bias).

“Signs of the dystrophic disease” should be clarified. The sentence has been clarified in the text.

Which muscles had a more oxidative phenotype at 3 weeks?

We agreed with the Reviewer that this was an interesting point to explore. Therefore, we performed Real Time PCR analysis of typical markers of the oxidative phenotype in different muscles at 3 weeks. As shown in the graphs below, all markers were significantly upregulated in *Sgca* null:*Nfix* null mice, in both slow-twitching and fast-twitching muscles. We decided to add the graphs in Fig. S2.

Real Time PCR showing expression of SDH-A/B and Cox5 in *Sgca* null and *Sgca* null:*Nfix* null muscles at 3 weeks. N=5 mice for each group. Mean±SD. *P<0,05; **P<0,01; ***P<0,001; ns: non significant.

Was there a change in the amount of myostatin protein in the muscles?

To answer the question raised by the Reviewer, we performed an ELISA assay detecting Myostatin protein in

skeletal muscle extracts. As shown in the graph below, the results confirmed transcript analysis, showing no change in the amount of Myostatin protein in muscles.

Myostatin Elisa assay on gastrocnemius muscles at 8 weeks; N=5 WT, N=8 *Sgca* null, N=8 *Sgca* null:*Nfix* null, N=5 mdx, N=4 mdx:*Nfix* null mice; mean±SD; ns: non significant.

What is meant by the “differentiation phase”? Does this refer to muscle development? If it means regeneration, this should be made clear. We substituted differentiation with regeneration in the text, as this was the intended meaning.

Suggests, not suggest We checked and corrected this in the text.

Page 10

Clarify “pass through” We substituted it with “is not mediated by”, to better clarify it in the text.

Delete “delay” unless there are data to support this. “Delay” was substituted with “improvement”.

“Already appeared” – does this mean that there was muscle pathology present before the intervention?

This sentence refers to electroporated mice. Since this experiment was conducted on mice at 5 weeks, we can state, based on Fig.1B showing muscle histology at 5 weeks, that the muscle pathology was already present before the intervention. Of course, it was not possible to check it on the same mice, but this refers to the general knowledge that at 5 weeks mice are already affected by the pathology.

Was the reduction in fibrosis statistically significant? To answer this question, we performed immunofluorescence for Collagen I, a marker of fibrotic areas, followed by quantification. Results demonstrated that the reduction in fibrosis is statistically significant, as shown in the following Figure that has now been inserted in the manuscript Fig. 7.

Quantification of Collagen I positive areas in *Sgca* null Tibialis anterior muscles electroporated with scramble (N=9) or shNfix (N=9) plasmids. **P<0,01.

Discussion
Page 11

What is meant by a “resolved” therapy? “Resolved” was substituted with “effective”.

Discuss the mouse soleus muscle –does this muscle have protection from oxidative damage? We better discussed this point in the Discussion.

“Persistent over time” should be deleted or qualified, as this was not a long-term investigation. “Persistent over time” was substituted with “persistent up to 6 months”, which was the last time point investigated.

Page 12

“Wide, striking amelioration” should be qualified. “Wide” and “striking” were deleted and substituted with significant, referring to the significant differences in terms of central nucleation, CSA, inflammation and running performances discussed throughout the manuscript.

Clarify what model mdx is less severe than. A clarification for that was added in the manuscript.

What is meant by “pushing on regeneration”? To clarify this concept, we substituted “pushing on regeneration” with “by promoting regeneration”.

Clarify the sentence beginning “Our results, beyond the...” This sentence was linked to the following one, which is more explicative.

Clarify the sentence beginning “The original idea...” By linking this sentence to the previous one, the concept is now much clearer.

Methods
Page 14

Why was a dough diet used? The dough diet was described in Campbell et al., 2008 as a way to increase the viability of the delicate *Nfix* null mouse model. To avoid any possible effect of the diet on the phenotype analysed, all the mice used for this study (including WT, *mdx* and *Sgca* null control mice) were fed with the same soft diet.

Are there any differences (in the literature) between male and female mice of the different models?

We found no described difference in the literature for *Sgca* null male and female mice. For *MDX* mice, we found the following two references highlighting a better-preserved specific tetanic force in female mice. However, no reference was found for a difference in terms of muscle histology, central nucleation or resistance to fatigue.

Moreover, for our experimental set up, mice were always compared to an age and sex-matched controls, to avoid possible differences.

Hourdé et al., 2013; *Protective effect of female gender-related factors on muscle force-generating capacity and fragility in the dystrophic mdx mouse*. Muscle Nerve. 2013 Jul;48(1):68-75. doi: 10.1002/mus.23700.

Hakim and Duan, 2012; *Gender differences in contractile and passive properties of mdx extensor digitorum longus muscle*. Muscle Nerve. 2012 Feb;45(2):250-6. doi: 10.1002/mus.22275.

Page 16

10 ml/g cannot be correct. We thank the Reviewer for this observation that is obviously a formatting mistake. We corrected this in the text, the right unit is $\mu\text{l/g}$.

Page 17

Sections, not slices. We corrected it in the text.

Figures

There is obvious ice crystal artefact in many of the sections (e.g. figure 2, top panels A and B, figure 4 top left panel B, Figure 5, top right panel A) –panels with ice crystal artifact should be replaced with images of muscles that do not have this artefact.

There seems to be an artefact in figure 6 C (knife cutting artefact?) –

Such images are not suitable for publication.

We agree with the reviewer and substituted the less suitable images as requested.

Figure 2 – “over time” should be qualified. Over time was substituted with “up to 6 months”, which is the latest time point analysed, now in new Fig. 2

Figures 3 and 4 – CSA data are better shown as graphs with size bins. We modified the graph type, in accordance with what suggested also by Reviewer 1, and added them to Fig.S1, Fig.S4 and Fig.7.

Figure 3

What was the % of EDB+ fibers in the TAs?

To answer this question, we quantified EBD+ fibers in the TAs of *Sgca* null and *Sgca null:Nfix* null mice at 8 weeks. As shown in the graph below (now displayed in the Manuscript in Fig. 3), also in the TAs there is a reduction of EBD+ fibers in dystrophic animals lacking *Nfix*.

Percentage of EBD positive myofibers in Tibialis anterior muscles at 8 weeks; N=19 *Sgca* null and 9 *Sgca null:Nfix* null mice; mean ± SD. *P<0,05.

Give data for collagen quantification. The quantification of collagen deposition is shown in Supplementary Fig.2.

Figure 4 D – why were so few WT fibers quantified? We increased the number of WT mice and counted more fibers (for a total of 799). The new graph is now shown below and in the main Figure.

Myofiber cross sectional area distribution at 8 weeks of age; N=799 fibers for WT, 908 for *Sgca* null and 1026 for *Sgca* null:*Mlc1f-Nfix2* mice.

Reviewer#4

For muscular dystrophy therapy, it has long been speculated that decreasing degeneration through promoting regeneration would be beneficial. However, none of the therapies based on this strategy have been successful. Previous work in this laboratory has shown that the Nfix transcription factor is responsible for the transition from embryonic to fetal myogenesis and is crucial for timing muscle regeneration post-injury. In the present study, this group has shown that dystrophic (*mdx* and *alpha-sarcoglycan*) Nfix-knockout mice display slowed regeneration and significantly improved muscle pathology and function. On the other hand, overexpression of Nfix (*alpha-sarcoglycan* null/*Mlc1f-2-Nfix2* mice) had a worsened phenotype, supporting the therapeutic value of Nfix downregulation. To understand how Nfix knockout improves the dystrophic phenotype, Rossi et al performed SDH staining and found that the transgenic muscle was more oxidative. While previously speculated, they were unable to find a link between Nfix and utrophin or myostatin expression. Finally, adult *alpha-sarcoglycan*-null mouse muscle was transfected with control or shNfix plasmids. Overall, muscle fibers were more normalized in area, with less regenerating fibers and inflammatory infiltration and less fibrosis. In summary, this study provided a convincing argument for Nfix downregulation therapy in muscular dystrophy as well as new insight for therapeutic strategies.

General comments, suggestions & concerns

- Would there be any long-term consequences of reduced regenerative capability of muscle?

We extended our analysis to 6-months of age (see Fig below, now also shown in Fig. 2 in the MS). As shown in the Figure, the amelioration of the dystrophic muscle was still evident up to 6 months in *Sgca* null:*Nfix* null mice, thus demonstrating that the effect due to absence of Nfix is persistent over time. Regarding the concern that the reviewer arose, what we are observing is not a defect of muscle regeneration but rather a delay of it. Therefore, we do not expect any long-term consequences, as regeneration anyway occurs in double null mice, as now confirmed by Fig. 4 in the new version of the Manuscript and in accordance with what we previously demonstrate in the context of acute damage (Rossi et al., Cell Reports 2016).

Haematoxylin and Eosin and Milligan's Trichrome stainings of *Tibialis anterior* and Diaphragm muscles from *Sgca* null and *Sgca* null:*Nfix* null mice at 6 months.

- Is there any link between *Nfix* and other cells involved in the dystrophic phenotype (ex. immune cells or fibroblasts)? Would *Nfix* have any effect on the activity of these cells?

We thank the Reviewer for this question. We observed that *Nfix* is also expressed by macrophages and for this reason we are developing a parallel project in the laboratory to identify the role of *Nfix* in macrophages. In any case, as also now more extensively discussed in the Discussion section of the manuscript, the exacerbation of the phenotype in the dystrophic mice overexpressing *Nfix* downstream a muscle specific promoter (*Mlc1f*) is clearly suggesting that function of *Nfix* in skeletal muscle cells is sufficient *per se* to mediate the phenotype. We clearly believe that a deeper investigation on the possible role of *Nfix* in immune cells would be both necessary and informative, but not in the present study because beyond the scope of the work.

- Did you perform any studies looking at fiber type in the adult treated muscle and what were the results?

We thank the Reviewer for this suggestion that we addressed thus improving our work. We performed SDH staining of treated scramble and sh*Nfix* muscles (by electroporation). As shown in the Figure below (now Fig. 7 of the MS), we observed a switch towards more oxidative SDH-positive dystrophic fibers upon silencing *Nfix*, thus reinforcing the main message of the work and its translational future application.

SDH staining of *Sgca* null *Tibialis anterior* muscles electroporated with scramble or sh*Nfix* vectors.

- Can you mention any proposed studies looking at the mechanism of action? Is the switch to an oxidative phenotype the only way *Nfix* knockout is working?

As discussed in the main text, and as now supported by regeneration experiments added to this last version of the Manuscript, we are proposing that *Nfix* exerts its functions through a double mechanism, involving both a switch to an oxidative phenotype and a delay of the regenerative process. We cannot absolutely exclude other mechanisms involved, since *Nfix* is also expressed by other cell populations such as the macrophages. Nevertheless, the exacerbated phenotype observed in the double *Sgca* null:*Mlc1f-Nfix2* mice, clearly support the evidence that a slow regenerating and a slow twitching dystrophic musculature are at the basis of the

amelioration/rescue observed. After all, even the soleus (a typical slow muscle) is itself affected in the dystrophic animals, suggesting that an oxidative phenotype is not sufficient to protect the musculature from the degeneration. This latter consideration also supports that even a delayed muscle regeneration is required, as also our results demonstrated.

- Overall, very interesting study and well-designed experiments.

REVIEWERS' COMMENTS:

Reviewer #1 (Remarks to the Author):

I believe the efforts made by the authors to address the reviewers' concerns were able to clarify a number of points and improve the manuscript. The authors addressed all my comments, with the only exception of the possibility to further assess the concept of delayed muscle regeneration, by performing muscle injury in *Sgca/Nfix* null mice and analyze the effects at 7 days post-injury. Actually they did not even comment about this point. Since I am quite convinced by the other pieces of evidence provided by the authors in this work, I still believe that their study would benefit from a further and solid proof-of-concept. Indeed, if regeneration is delayed, an injury model should display a worse phenotype in their mice. I guess the authors could claim, as they replied in response to another comment, that further studies could be in sharp contrast with the 3Rs principles for the use of animals for scientific purposes. While I am constantly trying to adhere at most to these ethics principles, I would have appreciated if the authors had addressed or at least commented this specific point.

Regarding the quantification of F4/80 positive macrophages in tibialis anterior (Fig. 3F), I believe that it should be referred to muscle area, rather than fiber ratio, since inflammation is not a process strictly related to the single fiber. In addition, a further muscle (e.g. diaphragm) could have been analyzed to strengthen the data and conclusions.

Minor comment: please correct "..quantification of macrophages is muscles." with "in" (page 13).

Reviewer #2 (Remarks to the Author):

The authors have replied to all my comments. Congratulations for a very original and very interesting paper that should be of a wide interest to readers in the muscle regeneration, muscular dystrophy, and therapeutic strategies fields

Reviewer #3 (Remarks to the Author):

This is a much improved manuscript and the authors have adequately addressed my queries.

There remain a few minor points:

A few typographical errors and errors with the English:

Page 6:

histopathological is misspelt

Page 8:

Sgca null:*Nfix* null mice, which start to massively regenerate

- Make clear that the muscles, not the mice, are regenerating.

Page 9:

slow-twitching and fast-twitching muscles -
twitch, not twitching

Figures:

There is still obvious ice crystal artefact in images of some sections, e.g. figure 2B left and middle panels and panel B and panel A of later figures. It is such a pity that the authors were not able to supply better images – this does detract from the quality of the paper.

REVIEWERS' COMMENTS:

Reviewer #1 (Remarks to the Author):

I believe the efforts made by the authors to address the reviewers' concerns were able to clarify a number of points and improve the manuscript. The authors addressed all my comments, with the only exception of the possibility to further assess the concept of delayed muscle regeneration, by performing muscle injury in *Sgca/Nfix* null mice and analyze the effects at 7 days post-injury. Actually they did not even comment about this point. Since I am quite convinced by the other pieces of evidence provided by the authors in this work, I still believe that their study would benefit from a further and solid proof-of-concept. Indeed, if regeneration is delayed, an injury model should display a worse phenotype in their mice. I guess the authors could claim, as they replied in response to another comment, that further studies could be in sharp contrast with the 3Rs principles for the use of animals for scientific purposes. While I am constantly trying to adhere at most to these ethics principles, I would have appreciated if the authors had addressed or at least commented this specific point.

We thank the Reviewer for her/his comments and acknowledgments.

Regarding her/his still open concern, we would like to point Reviewer's attention to the fact that we already demonstrated, in a CTX injury model and the following analysis of muscle regeneration at different time points (including the suggested time point 7), that mice lacking *Nfix* have an impairment in the regeneration process, which happens correctly but is slower if compared to wild types (Rossi et al., Cell Reports 2016). We therefore expect that, if we repeat the same experiment in a dystrophic context, results would be in line with that. This evidence is also supported by the analysis of regenerating (dMyHC+) fibers in dystrophic mice lacking *Nfix*, that we added to the revised version of the manuscript. Indeed, one possible limitation of the experimental set up proposed by the Reviewer is the impossibility to check muscle regeneration and dMHC expression before the CTX damage. Without knowing the basal level of expression of this marker before injury for each sample, and considering that dystrophic mice (independently from *Nfix*) already show a certain degree of variability in terms of regeneration at different time points, it would be very difficult to discriminate between the effects of the chronic versus the acute injury. Indeed, regeneration already occurs in dystrophic mice, even in the absence of CTX treatment.

This is not because we do not want to perform this kind of experiment, but rather because we do not envisage the best and convincing way to develop and evaluate the possible results.

We therefore think that the analysis of endogenous dMyHC expression over time in dystrophic mice with or without *Nfix*, that we added to the revised version of the Ms, is the cleanest way to demonstrate that regeneration is delayed in dystrophic mice lacking *Nfix*.

Regarding the quantification of F4/80 positive macrophages in tibialis anterior (Fig. 3F), I believe that it should be referred to muscle area, rather than fiber ratio, since

inflammation is not a process strictly related to the single fiber. In addition, a further muscle (e.g. diaphragm) could have been analyzed to strengthen the data and conclusions.

We thank the Reviewer for the suggestion. We modified the graph accordingly (now in Fig 3F).

Minor comment: please correct “..quantification of macrophages is muscles.” with “in” (page 13).

We corrected the typo in the text.

Reviewer #2 (Remarks to the Author):

The authors have replied to all my comments. Congratulations for a very original and very interesting paper that should be of a wide interest to readers in the muscle regeneration, muscular dystrophy, and therapeutic strategies fields

Reviewer #3 (Remarks to the Author):

This is a much improved manuscript and the authors have adequately addressed my queries.

There remain a few minor points:
A few typographical errors and errors with the English:
Page 6: histopathological is misspelt

We corrected the typo in the text.

Page 8:
Sgca null:Nfix null mice, which start to massively regenerate
- Make clear that the muscles, not the mice, are regenerating.

We corrected the sentence in the text.

Page 9:

slow-twitching and fast-twitching muscles - twitch, not twitching

We corrected the typo in the text.

Figures:

There is still obvious ice crystal artefact in images of some sections, e.g. figure 2B left and middle panels and panel B and panel A of later figures. It is such a pity that the authors were not able to supply better images – this does detract from the quality of the paper.

We thank the Reviewer for the suggestion. As explained in the first answer, we substituted images with obvious crystal artefacts. We believe that the image quality is now suitable for publication. Minor artefacts that may still be present could not be avoided due to the applied inclusion procedure.